# Distributed subthreshold representation of sharp wave-ripples by hilar mossy cells

**Ayako Ouchi[1,2]\*, Taro Toyoizumi[3,4], Nobuyoshi Matsumoto[1,5], Yuji Ikegaya[1,5,6]**

[1]Graduate School of Pharmaceutical Sciences, The University of Tokyo, Tokyo, Japan; [2]Laboratory for Systems Neurophysiology, RIKEN Center for Brain Science, Wako, Japan; [3]Laboratory for Neural Computation and Adaptation, RIKEN Center for Brain Science, Wako, Japan; [4]Department of Mathematical Informatics, Graduate School of Information Science and Technology, The University of Tokyo, Tokyo, Japan; [5]Institute of AI and Beyond, The University of Tokyo, Tokyo, Japan; [6]Center for Information and Neural Networks, National Institute of Information and Communications Technology, Suita, Japan

**Abstract** In neural information processing, the nervous system transmits neuronal activity between layers of neural circuits, occasionally passing through small layers composed only of sparse neurons. Hippocampal hilar mossy cells (MCs) constitute such a typical bottleneck layer. However, how efficient information encoding is achieved within such constrained layers remains poorly understood. To address this, we focused on sharp wave-ripples (SWRs) – synchronous neural events originating in the CA3 region – and investigated functional diversity within MC populations using in vivo/in vitro patch-clamp recordings in mice. By combining machine learning algorithms, we developed a model to predict CA3 SWR waveforms based on the synaptic response waveforms of MCs, suggesting that SWR-related information is indeed encoded in their subthreshold activity. While individual MCs were generally associated with specific SWR clusters, partial overlap across some MCs was also observed, indicating that CA3 activity is distributed across the MC population. Our findings suggest that CA3 SWR activity is represented in a pseudo-orthogonal manner across MC populations, allowing the small MC layer to effectively compress and relay hippocampal information.

**\*For correspondence:**
ayako.ouchi515@gmail.com

**Competing interest:** The authors declare that no competing interests exist.

## Editor's evaluation

The study shows that activity of dentate gyrus mossy cells encode information from sharp wave-ripple complexes (SWRs) from the adjacent CA3 region. The study used difficult methods such as recording from multiple mossy cells simultaneously, as well as deep learning, which is impressive. The findings are fundamental in significance because they show a relationship between mossy cells and sharp wave ripples that has not been appreciated before, and the strength of evidence is compelling.

## Introduction

Neural information is processed as it travels through the various layers of the neural circuit, with the number of neurons in each layer varying from layer to layer. In general, a layer with a large number of neurons is assumed to be able to process larger and more complex information, while a layer with a small number of neurons must compress or otherwise trim the information. Neural processing with a small layer may be advantageous for dimensionality and feature extraction of information (*Grezl et al., 2007*).

Mossy cells (MCs) in the hippocampal formation serve as a good experimental model for such a bottleneck middle layer (*Figure 1—figure supplement 1*). They are the only excitatory neurons that exist in the dentate hilus, a small zone intercalated between the CA3 region and the dentate gyrus (DG). MCs receive synaptic inputs from CA3 pyramidal cells (*Scharfman, 1996*; *Scharfman, 1994a*), and their axons send synaptic outputs to DG granule cells (*Buckmaster et al., 1992*). While it has been anatomically shown that axons from CA3 pyramidal cells project to the inner molecular layer and granule cell layer, it is not clear whether they actually form synapses and serve their function (*Li et al., 1994*; *Vivar et al., 2012*). Thus, MCs are thought to relay neuronal activity from the CA3 region to the DG. An estimated 30,000 MCs reside in the rat hippocampal formation, whereas approximately 300,000 and 1,000,000 CA3 pyramidal and granule cells exist, respectively (*Amaral et al., 1990*; *Jinno and Kosaka, 2010*). Thus, the number of MCs is one to two orders of magnitude lower than that of CA3 pyramidal neurons and DG granule cells (*Figure 1—figure supplement 1*), and MCs constitute a small relay layer.

Sharp wave-ripples (SWRs) consist of two distinct components: the sharp wave and the ripple. The sharp wave is a slow deflection in the local field potential (LFP), primarily originating from the synchronous burst firing of CA3 pyramidal neurons. The ripple is a high-frequency oscillation (80–200 Hz) that is most prominently observed in the CA1 region, though it is also present in the CA3 region. The ripple component is superimposed on the sharp wave. SWRs primarily occur during sleep or quiet wakefulness (*Buzsáki et al., 1992*) and are thought to represent a mechanism for reactivating or consolidating recent experiences. These 'replayed' neural events allow information to be transferred from the CA3 region forward to the neocortex via the subiculum (*Nitzan et al., 2020*; *Siapas and Wilson, 1998*; *Sirota et al., 2003*) or to the deep layers of the entorhinal cortex (*Chrobak and Buzsáki, 1994*). SWRs may also propagate backward to the DG (*Penttonen et al., 1997*; *Scharfman, 2007*). In this case, SWRs generated by the population activity of CA3 pyramidal cells may propagate to the DG via synaptic inputs to MCs, which in turn provide excitatory input to GCs, forming a potential disynaptic pathway. The observation of delayed synaptic responses in GCs compared to MCs during SWRs supports the hypothesis that the CA3-DG backpropagation reflects SWR-associated excitatory activity mediated through MCs (*Scharfman, 1994a*; *Scharfman, 1994b*; *Scharfman, 1996*; *Scharfman, 2007*; *Ouchi et al., 2017*; *Swaminathan et al., 2018*). Furthermore, both experimental and theoretical studies have demonstrated that the MC activity can be involved in memory processes such as spatial coding and pattern separation (*Danielson et al., 2017*; *GoodSmith et al., 2017*; *Senzai and Buzsáki, 2017*; *Myers and Scharfman, 2011*). In addition, MC synaptic activity serves as temporary storage and encoding of information at the local circuit level (*Hyde and Strowbridge, 2012*).

During SWRs, spatiotemporal patterns of neuronal firing in the hippocampus that resemble activity during an experience are replayed as rapid spike sequences (*Diba and Buzsáki, 2007*; *Lee and Wilson, 2002*; *Skaggs and McNaughton, 1996*; *Wilson and McNaughton, 1994*). Different SWRs may reactivate different sequences of spikes, possibly encoding information that correlates with different experiences. Neural information in SWRs is manifested in the LFP waveforms of SWRs (*Navas-Olive et al., 2022*; *Taxidis et al., 2015*). Accordingly, the characteristics of SWR waveforms, including their amplitudes and widths, exhibit heterogeneity across individual SWRs. Therefore, comparing the SWR waveforms with SWR-induced depolarizations in MCs provides us with a unique opportunity to investigate how neural information travels through a small layer.

In the present work, we performed patch-clamp recording from up to five MCs together with recording of CA3 SWRs and developed a neural network that associates SWRs with membrane potential ($V$m) dynamics in MCs. Our neural network was able to predict SWR waveforms from MC $V$ms. Strikingly, about 9% of the total SWR waveform could be predicted from even a single MC. Neural representations of MCs cover a wide range of the SWR repertoire with slight overlap between MCs, and as a result, the proportion of predictable SWRs increased sublinearly as the number of MCs used for prediction increased.

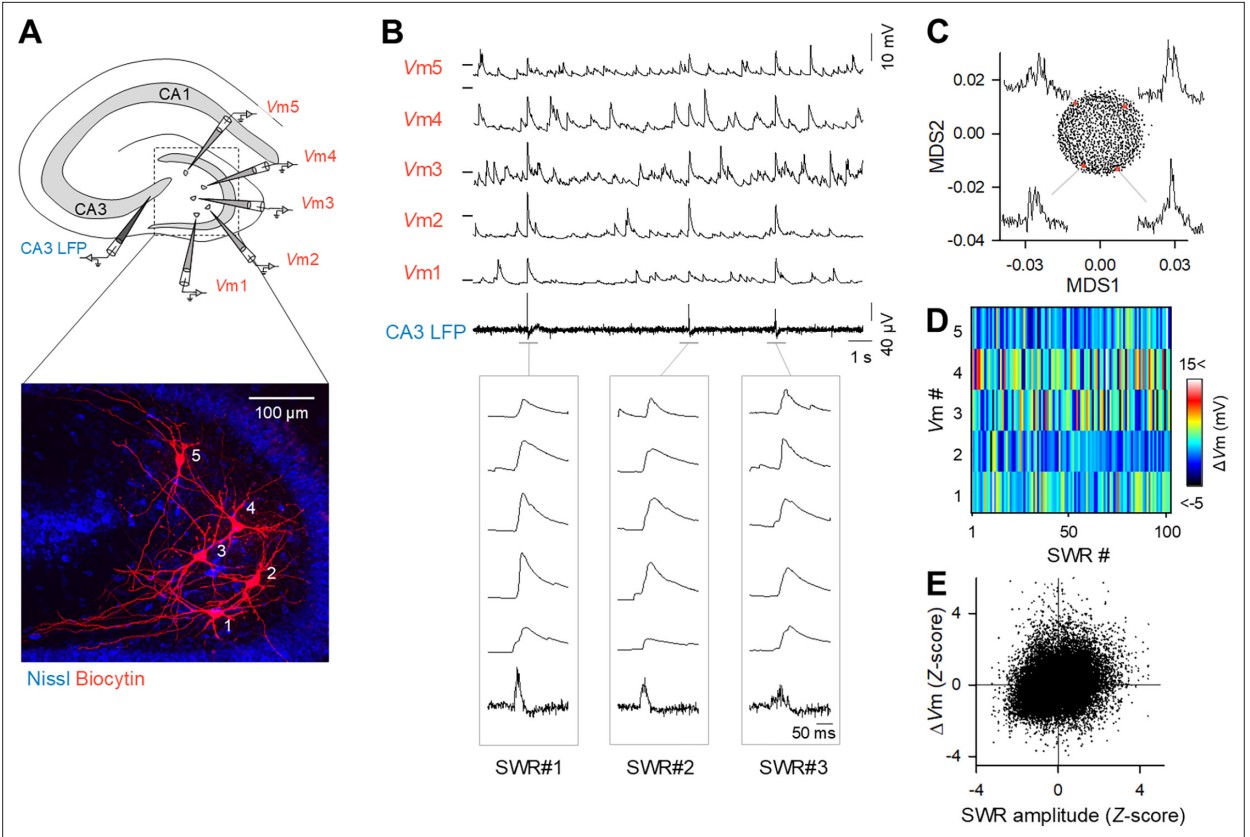

**Figure 1.** Spatiotemporal diversity of CA3 sharp wave-ripple (SWR)-induced ΔVms in mossy cells (MCs). (**A**) (Top) Schematic illustrations of whole-cell Vm recordings from five MCs together with local field potential (LFP) recordings from the CA3 region. (Bottom) Recorded cells were confocally identified by immunostaining for intracellularly injected biocytin (red) in a Nissl-counterstained slice (blue). (**B**) Representative traces of five MC Vms and CA3 LFPs. The horizontal ticks on the left of the Vm traces represent –60 mV. The traces during the three SWRs are expanded in time in the bottom boxes. (**C**) The LFP traces of 945 SWRs in a representative dataset were dimensionally reduced to two dimensions using the multidimensional scaling (MDS) algorithm based on the root mean square errors (RMSEs) (i.e. dissimilarity) between each SWR waveform. Each dot indicates a single SWR event. The SWR waveforms connected to the red dots correspond, indicating that the SWR waveforms differ depending on the location of the MDS plot. That is, visually similar SWRs are plotted proximally by the MDS algorithm. (**D**) A pseudocolor map for the amplitudes of the SWR-induced depolarizations (ΔVm) in five MCs during a total of 102 SWRs, demonstrating the heterogeneity of Vm responses among MCs across SWRs. (**E**) Weak correlations between the SWR amplitudes and ΔVms of MCs. Each dot indicates a single SWR event. ΔVms and SWR amplitudes were Z-standardized for each recording electrode before the data were pooled. $R=0.25$, $n=28,463$ SWR events in 87 mossy cells in 23 slices, from 14 mice.

The online version of this article includes the following figure supplement(s) for figure 1:

**Figure supplement 1.** Estimated total number of excitatory neurons in each subregion in the rat hippocampus.

**Figure supplement 2.** Sharp wave-ripple (SWR) onset-triggered average of ΔVm of mossy cells (MCs).

**Figure supplement 3.** Lack of sharp wave-ripples (SWR)-induced depolarization of mossy cells (MCs) isolated from the CA3 region.

**Figure supplement 4.** Sharp wave-ripples (SWR) frequency does not vary during recording time.

## Results

### Simultaneous patch-clamp recordings from hilar MCs in vitro and in vivo

Using mouse acute hippocampal slice preparations, we simultaneously recorded Vm from multiple neurons in the hilus in a whole-cell current-clamp configuration, as well as LFPs from the CA3 pyramidal cell layer (*Figure 1A*). A post hoc biocytin-based visualization method was used to confocally image the recorded neurons in NeuroTrace Nissl-counterstained slices (*Figure 1A*, inset). As a result, we obtained 2 quintuple, 8 quadruple, and 15 triple patch-clamp recordings from MCs in 23 slices from 14 mice. The recording periods ranged from 56 to 357 s (median = 301 s, *Supplementary file 1*).

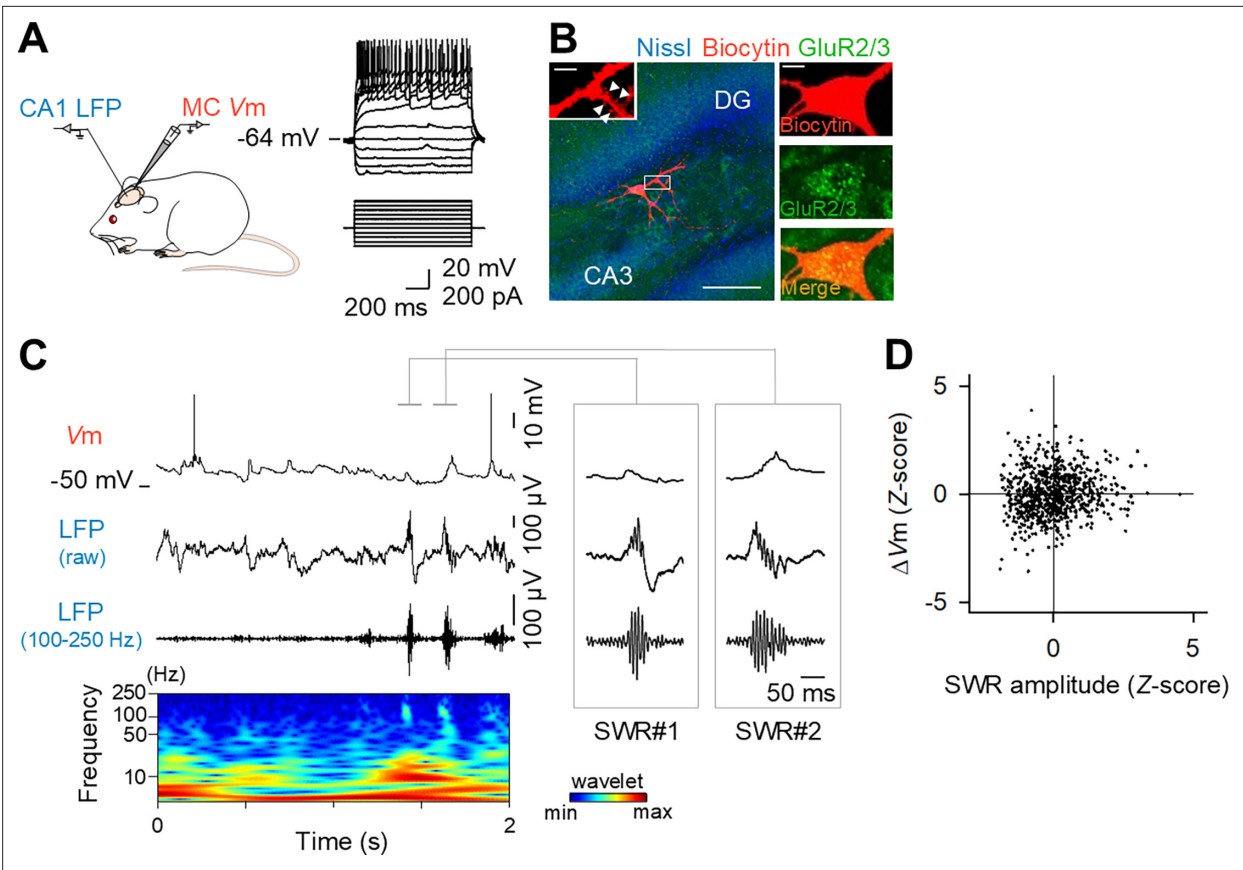

**Figure 2.** Diversity in sharp wave-ripple (SWR)-induced mossy cell (MC) ΔVms in vivo. (**A**) MCs were current-clamped in a urethane-anesthetized mouse, while the local field potentials (LFPs) were recorded from the CA1 region. (**B**) The recorded neurons were confocally visualized by the detection of intracellularly loaded biocytin (red) in a section immunostained with anti-GluR2/3 (green), an MC marker, and counterstained with NeuroTrace Nissl (blue). Scale bar: 100 μm. The high-magnification images on the right indicate that the recorded neuron was positive for GluR2/3 and that thorny excrescences were observed on proximal dendrites. Scale bar: 10 μm. (**C**) (Top) Representative Vm traces of an MC. (Middle) A raw LFP trace recorded simultaneously from the CA1 region and its bandpass-filtered trace (100–250 Hz). The bottom panel indicates the wavelet spectrogram of the raw LFP trace. The traces during two SWRs are expanded in time in the right insets. (**D**) Weak correlations between the SWR amplitudes and ΔVms of MCs during SWRs. R=0.1053, n=756 SWR events in six MCs, from 6 mice.

All tested slices spontaneously emitted SWRs in CA3 LFPs (*Figure 1B*). The frequency of SWR events ranged from 11 to 84 per minute (median = 38 per minute). The waveforms of SWRs were not uniform; e.g., their amplitudes and widths varied from SWR to SWR. Dimension reduction using the multidimensional scaling (MDS) algorithm revealed that the SWR waveforms formed a continuous set, because they were nondiscretely distributed in two-dimensional MDS space (*Figure 1C*).

MCs reliably exhibited depolarization immediately after SWR onset (*Figure 1B*, *Figure 1—figure supplement 2*). This reliability is in marked contrast to that of the CA1 pyramidal cells and DG granule cells, which do not always depolarize for all SWRs (*Mizunuma et al., 2014*; *Ouchi et al., 2017*). The Vm responses of MCs were abolished when the dentate hilus was surgically isolated from the CA3 region (*Figure 1—figure supplement 3*), indicating that these responses were caused by CA3 SWR activity.

Although the SWR frequency does not vary during the recording time (*Figure 1—figure supplement 4*), the Vm responses of MCs to SWRs were also heterogeneous; a single MC exhibited different depolarization amplitudes (ΔVms) for different SWR events, whereas a single SWR induced different ΔVms in different MCs (*Figure 1D*). Overall, the ΔVms in MCs were only weakly correlated with the amplitude of SWRs (*Figure 1E*; R=0.25, n=28,463 SWRs from 87 cells), indicating that even a small SWR could cause a large ΔVm in an MC and vice versa.

We also performed in vivo whole-cell recordings from MCs in urethane-anesthetized mice (*Figure 2A*). The recorded cells were identified based on their morphological features and

immunoreactivity for glutamate receptor 2/3 (GluR2/3), a molecular marker of MCs (*Figure 2B*). As observed in vitro, MCs exhibited diverse Δ*V*m responses to SWRs (*Figure 2C*), and again, Δ*V*ms correlated only weakly with SWR amplitudes (*Figure 2D*; *R*=0.1053, *n*=756 SWRs from 6 cells, 6 mice). Previous research has shown that the hyperpolarization of MC membrane potential associated with SWR indicates that SWR is related to the inhibition of MCs (*Henze and Buzsáki, 2007*; *Soltesz et al., 1993*). Our data showed that the proportion of cases of depolarization or hyperpolarization was about the same, with a slight excess of depolarization. It should be noted that MCs are highly active and fluctuating cells, and the determination of whether they are depolarized or hyperpolarized is highly dependent on the method of analysis. Moreover, the firing rate of MCs during the recording period was 1.07±0.93 Hz (mean ± SD from 6 cells, 6 mice), whereas the proportion of MC firing activity recruited during all SWRs was 6.68 ± 4.79% (mean ± SD from 6 cells, 6 mice, *n*=757 SWR events; calculated as firing within 50 ms after the SWR peak).

The weak correlations between Δ*V*ms and SWR amplitudes suggest that MCs respond to SWRs in a complex, rather than a simple linear, manner. The variability in Δ*V*ms responses suggests that a population of MCs may collectively cover information of diverse SWRs. Therefore, we hypothesized that, if this is the case, SWR waveforms (SWR-related information) could be predicted from the *V*m responses of a set of MCs.

## Machine learning-based prediction of SWR traces from *V*ms in MCs

To associate MC *V*ms with SWR waveforms, we trained a neural network model with a hidden layer (*Figure 3A*). For each SWR event in quintuple patch-clamp recordings, we extracted 100 ms segments of the *V*m and LFP traces between –50 and +50 ms relative to the SWR peak times. We divided these segments into five subsets and trained the neural network using *V*m and LFP traces in the four out of five subsets (80%) to predict the SWR waveforms in the remaining subset (20%) that was not used for training (*Figure 3B*). We repeated this procedure so that all SWRs were targeted for prediction. For each prediction, the prediction error was quantified by the root mean square error (RMSE) between the original and predicted SWR waveforms. We also trained the same neural network using surrogate data in which the combinations of *V*ms and LFPs were randomly shuffled across SWR events within each cell. We then compared the RMSEs for the LFP waveforms predicted by the original and shuffled *V*m data. The RMSE produced from predictions using real data was significantly lower than those produced from shuffled data (*Figure 3C*; p=2.11 × 10$^{-9}$), indicating that the neural network predicted the SWR waveforms from the *V*m dynamics of five MCs with significantly higher accuracy than chance. In other words, CA3 information during SWRs was at least partially preserved in the *V*m responses of MCs.

We next investigated the relationship between the number of MCs used to train the neural network and the prediction performance. To increase the number of datasets, we divided the quintuple, quadruple, and triple patch-clamp recording datasets into subsets with fewer MCs. The RMSEs calculated between the original and predicted traces, which were *Z*-standardized for each dataset, decreased as the number of simultaneously recorded cells used for prediction increased (*Figure 3D*). The predictive performance of RMSE predicted from real data and shuffled data according to the number of MCs was also evaluated using the *t* value of a paired t-test (*Figure 3E*). As another index of prediction accuracy, we calculated the wavelet coherence for a high-frequency range (120–250 Hz) between the original and predicted waveforms and found that the high-frequency details of the SWR waveforms predicted by a larger number of MCs were also closer to the original (*Figure 3—figure supplement 1*), suggesting that a greater number of MCs can retain more information about SWRs. Remarkably, the *V*ms of even a single MC were sufficient to significantly predict the SWR waveforms (*Figure 3E*, *Figure 3—figure supplement 2*; *t*=13.0646, p=6.79 × 10$^{-39}$, *n*=87 cells). We also trained the neural network using *V*m data obtained by in vivo single-cell patch-clamp recordings to predict the LFP waveforms during SWRs (*Figure 3—figure supplement 3*; *Figure 3F*). Again, the RMSEs predicted by the real datasets were significantly smaller than those predicted by the shuffled datasets (*Figure 3E*, *t*=2.2619, p=0.024, *n*=6 cells).

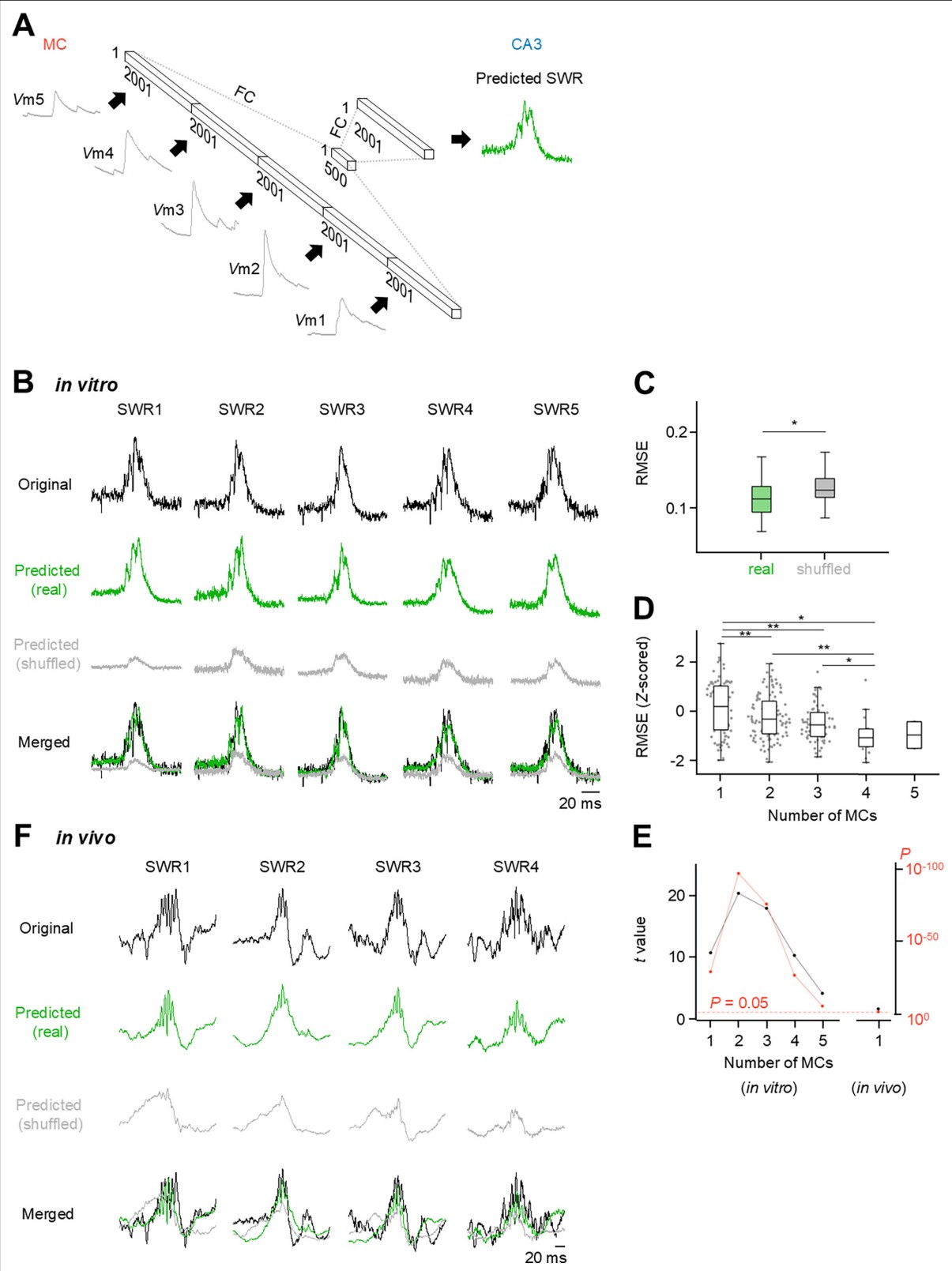

**Figure 3.** Machine learning-based prediction of sharp wave-ripple (SWR) traces from *V*ms in mossy cells (MCs) in vitro/in vivo. (**A**) Design of the neural network, which consisted of five layers connected unidirectionally with fully connected (FC) links. The number of *V*ms (left input) varies between 1 and 5, depending on the number of MCs. In this neural network, the SWR and the corresponding *V*m of MCs are linked and trained. Thus, by using the *V*m of the MC as input, the SWR waveform is obtained as output. (**B**) Five examples of original SWR traces (black) and SWR traces predicted from real

*Figure 3 continued on next page*

*Figure 3 continued*

*V*m datasets (green) and shuffled *V*m datasets (gray). All traces are merged at the bottom. Note that the vertical axis is scaled for each waveform. (**C**) The results show that the difference between RMSE$_{real}$ and RMSE$_{surrogate}$ is statistically significantly different from 0. RMSE$_{surrogate}$ was calculated as the average of the waveforms that were shuffled 100 times for each SWR. *n*=178 SWRs from one recording dataset of MC quintets, *p = 2.11 ×10$^{-9}$, paired t-test. (**D**) Root mean square error (RMSE) calculated between the original and predicted traces was decreased as the number of simultaneously recorded cells used for prediction increased. One cell: *n*=28,463 SWRs from 87 single MC recordings; 2 cells: *n*=35,914 SWRs from 113 MC pair recordings; 3 cells: *n*=19,874 SWRs from 67 trios; 4 cells: *n*=4,403 SWRs from 18 quartets; 5 cells: *n*=271 SWRs from 2 quintets, *F*=8.28, p=2.51 × 10$^{-6}$, one-way ANOVA. *p<0.05, **p<0.005, Fisher's LSD test. (**E**) The *t* values determined by the paired t-test are plotted in *black*, and the p-values for the corresponding *t* values are plotted in *red*, indicating that the *V*ms of one to five MCs led to a significantly accurate prediction. The *t* value and p-value were also calculated from in vivo dataset as in **C** and plotted on the right. (In vitro) One cell: *n*=28,463 SWRs from 87 single MC recordings; 2 cells: *n*=35,914 SWRs from 113 MC pair recordings; 3 cells: *n*=19,874 SWRs from 67 trios; 4 cells: *n*=4,403 SWRs from 18 quartets; 5 cells: *n*=273 SWRs from 2 quintets. (In vivo) *n*=757 SWRs from 6 mice, *p=0.024, *t*=2.2619, paired t-test. (**F**) Four examples of original SWR traces (black) and SWR traces predicted from real *V*m datasets (green) and shuffled *V*m datasets (gray). All traces are merged in the bottom panel. Note that the vertical axis is scaled for each waveform.

The online version of this article includes the following figure supplement(s) for figure 3:

**Figure supplement 1.** Reproducibility of 120–250 Hz coherence increases as the number of mossy cells (MCs) increases.

**Figure supplement 2.** Differences in prediction accuracy for each mossy cell (MC), slice, and animal.

**Figure supplement 3.** Design of in vivo neural network model.

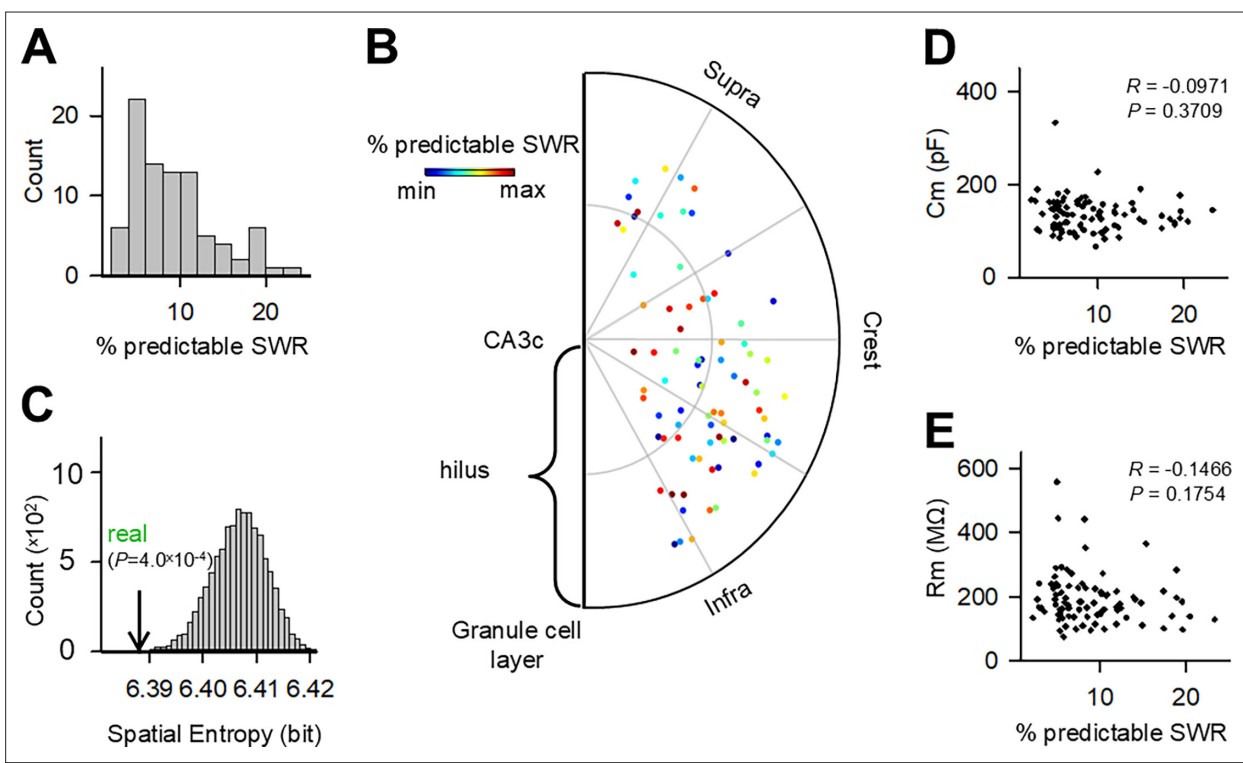

**Figure 4.** Correlation between proportions of predictable sharp wave-ripple (SWR) by single mossy cell (MC) and spatial distribution or fundamental neuronal properties. (**A**) Distribution of the % predictable SWR from 87 MCs. The % predictable SWR was computed from root mean square errors (RMSEs) by using a single MC prediction. (**B**) The relative locations of MCs in the hilus are plotted in a semicircular diagram. Each dot indicates a MC, and its color represents the % predictable SWR. (**C**) Spatial bias of (**B**) was evaluated by spatial entropy and compared to its chance distribution. The real entropy value is indicated by an arrow. (**D, E**) Correlations between % predictable SWR and electrophysiological properties. Each dot indicates a single MC. Cm: p=0.3709, *R*=−0.0971; Rm: p=0.1754, *R*=−0.1466, *n*=87 MCs.

The online version of this article includes the following figure supplement(s) for figure 4:

**Figure supplement 1.** Correlations between % predictable sharp wave-ripple (SWR) and mossy cell (MC) EPSPs.

## Proportion of predictable SWRs by single MC and correlation with spatial distribution or fundamental neuronal properties of MCs

Given that the SWR waveform was predicted with statistical significance even with a single MC, we examined the percentage of SWRs that a single MC could predict (predictable SWRs) relative to the total SWRs. For each MC, the significance of the RMSEs in predicting individual SWR waveforms was statistically determined (*Figure 4A*). Specifically, if the RMSE predicted by the real datasets (RMSE-real) was significantly lower than that predicted by the shuffled datasets (RMSE$_{surrogate}$), where RMSE$_{real}$ and RMSE$_{surrogate}$ represent the RMSEs between the original SWR waveform and the SWR waveforms predicted by the neural networks trained with the real *V*ms of the MC and its surrogate *V*ms, respectively, then the SWR was defined as 'predictable' by the MC. The surrogate *V*ms were generated by randomly shuffling *V*ms across SWRs within the MC. The percentage of predictable SWRs to all SWRs recorded from the MC was 9.2 ± 4.8% (mean ± SD of 87 MCs, median = 8.01). The relative positions of individual MCs in the dentate hilus were plotted on a standardized semicircular map, with the predictable SWR ratios of the corresponding MCs displayed in pseudocolor (*Figure 4B*; *Koyama et al., 2012*). We computed the spatial entropy of this distribution to examine whether MCs with specific SWR predictability were spatially clustered in a particular area. The spatial entropy was 6.39, and this value was significantly lower than the 95% confidence interval of a chance distribution computed from 10,000 proxy datasets in which the percentage of predictable SWRs was randomly shuffled across MCs without changing the positions of the MCs (*Figure 4C*, p=4.0 × 10$^{-4}$). Therefore, the anatomical locations of MCs were linked with their percentage of the predictable SWRs; specifically, the lower blade of the DG included MCs with higher predictability. However, neither the membrane capacitances (*Figure 4D*) nor the membrane resistances (*Figure 4E*) of MCs were correlated with the percentage of predictable SWRs of the MCs. Thus, the intrinsic electrophysiological properties of MCs were unlikely to determine their SWR predictability. We also examined the correlation between percentage of predictable SWR and EPSP of MCs (*Figure 4—figure supplement 1*). We found no correlation between EPSP amplitude and SWR, but a negative correlation between EPSP frequency and SWR, suggesting that the higher the activity of the MC, the more difficult it is to predict SWR, and that less active MCs predict more accurate SWR.

## Biased prediction of SWRs by MCs

We investigated whether SWRs with a given waveform were more predictable by a given MC. The dimensionality of the SWR waveforms was reduced to two dimensions using the MDS algorithm. Because the MDS algorithm preserves the relative distances (here, RMSE values) between any two SWRs, the proximity of SWR pairs in MDS space reflects the similarity of their original waveforms. As an example, we plotted a total of 945 SWRs recorded from a hippocampal slice in which four MCs were simultaneously patch-clamped (*Figure 5A*, left). The predictable SWRs are shown in red in the MDS space in *Figure 5A*. We then computed the spatial entropy of the predictable SWRs in the MDS plot and compared it to the chance distribution obtained by 1,000,000 surrogates in which the binarized prediction (i.e. predictable or non-predictable) was permuted across SWRs within the cell (*Figure 5A*, right). We repeated this comparison for a total of 87 MCs and found that for 37 out of 87 MCs, the spatial entropy was significantly lower than the chance value (*Figure 5B*). Thus, as a whole, a synaptic input to the MC received from the upstream network tended to predict a specific subset of SWRs with similar waveforms.

We examined how different MCs in a hippocampal slice predicted the different SWR subsets. For this purpose, we considered the binarized prediction scores for SWRs as a vector. Specifically, with 'predictable' as 1 and 'non-predictable' as 0, we created a binary vector for all SWRs in a given MC. We then calculated the correlation coefficient between the vectors of two MCs within each dataset. Data were pooled from a total of 113 MC pairs from 2 quintuple, 8 quadruple, and 15 triple recording datasets. The correlation coefficients varied mainly between 0 and 0.3, and their mean (0.054) was relatively small with SD = 0.105. Indeed, they did not differ from 0 as a whole distribution (Z=–0.002, p=0.998, *Z* test for single value comparison), indicating that a subset of SWRs that can be predicted by a given MC is nearly independent of that predicted by the other MCs (*Figure 5C*). Therefore, information in CA3 SWRs is represented in a manner that is nearly mutually exclusive (i.e. pseudo-orthogonal) among individual MCs.

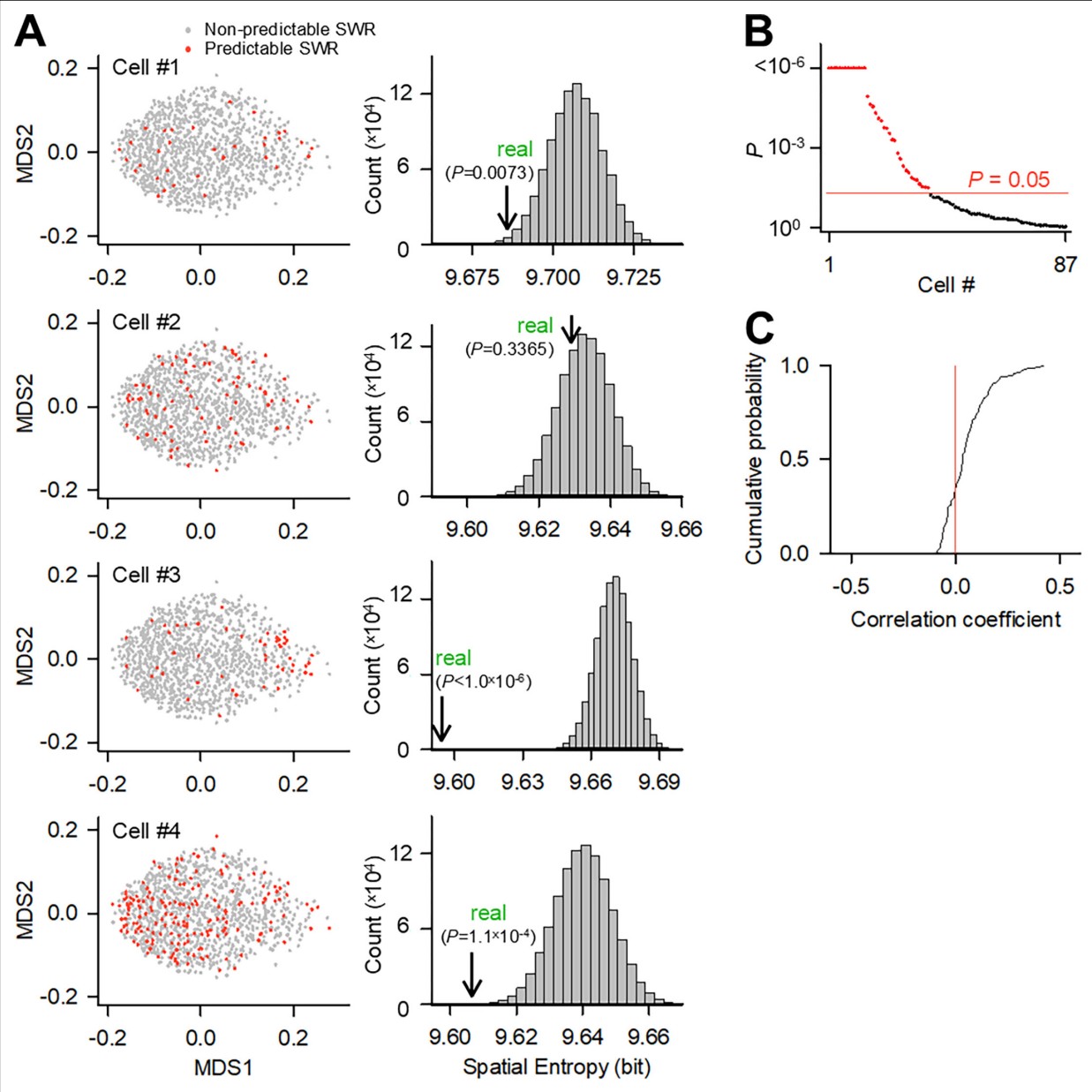

**Figure 5.** Biased prediction of sharp wave-ripples (SWRs) by mossy cells (MCs). (**A**) (Left) The local field potential (LFP) traces of 945 SWRs from a representative quadruple recording dataset were dimensionally reduced using the multidimensional scaling (MDS) algorithm. Each dot indicates a single SWR event, and its color represents whether the prediction rate is significant (red) or not (gray). (Right) The spatial bias of the prediction rates in the MDS space was evaluated by spatial entropy. Lower entropies indicate higher spatial biases of the prediction rate. In each MC, the entropy was compared to the change distribution obtained by 1,000,000 surrogates in which the prediction rates were shuffled within the MC. The real entropy value is indicated by an arrow with its p-value. (**B**) The same calculations as those in (**A**) were repeated for all 87 cells, and their p-values were plotted. The red dots indicate MCs with significantly lower entropy values. (**C**) Distribution of the correlation coefficient of the root mean square error (RMSE) score sets (non-predictable SWR as 0, predictable SWR as 1) between 113 MC pairs, which were obtained from 2 quintuple, 8 quadruple, and 15 triple recording datasets.

## Prediction specificity of SWRs by MCs

Further analysis was performed to confirm the pseudo-orthogonality of the information code. Binary vectors were computed in the analysis of *Figure 5A*. The vectors for five representative MCs recorded from the same hippocampal slice containing a total of 178 SWRs are shown in *Figure 6A*, indicating that sometimes different MCs were able to predict the same wave, sometimes not. The Venn diagram shows how the SWRs predicted by these five MCs overlapped (*Figure 6B*). The number of SWRs in

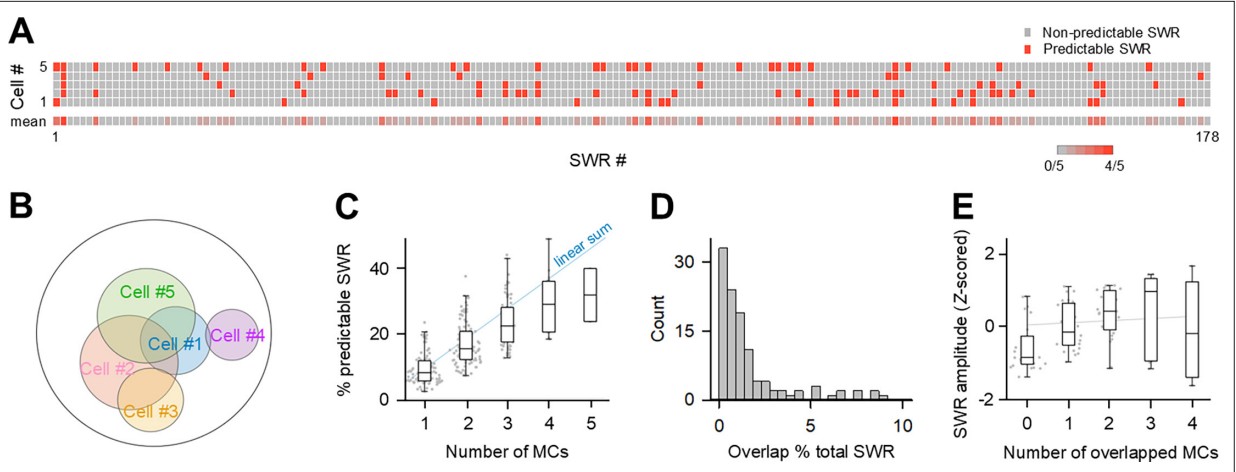

**Figure 6.** Prediction specificity of sharp wave-ripples (SWRs) by mossy cells (MCs). (**A**) The representative quintuple recording in a total of 178 SWRs. The top five rows demonstrating the SWRs that each MC can predict. Its color represents whether the SWR was predictable (red) or not predictable (gray). The mean predictivity across five MCs is shown in the bottom rastergram. (**B**) The Venn diagram demonstrated the image of the range of SWRs that each MC can predict out of the recorded SWR. (**C**) The % predictable SWR was increased as the number of simultaneously recorded cells used for prediction increased. The linear summation is indicated by a solid line (blue). One cell: $n$=28,463 SWRs from 87 single MC recordings; 2 cells: $n$=35,914 SWRs from 113 MC pair recordings; 3 cells: $n$=19,874 SWRs from 67 trios; 4 cells: $n$=4,403 SWRs from 18 quartets; 5 cells: $n$=271 SWRs from 2 quintets, *$p$=1.00 × 10$^{-38}$, Jonckheere-Terpstra test. (**D**) Distribution of the overlap % per total SWR between 113 MC pairs, which were obtained from 2 quintuple, 8 quadruple, and 15 triple recording datasets. (**E**) Correlations between SWR amplitudes and number of overlapped MCs. The regression line is indicated by solid line (gray). The data were obtained from 2 quintuple, 8 quadruple, and 15 triple recording datasets, $F$=5.38, *$p$=0.0007, one-way ANOVA; *$p$=1.59 × 10$^{-5}$, Jonckheere-Terpstra test.

The online version of this article includes the following figure supplement(s) for figure 6:

**Figure supplement 1.** Visualization of overlapped sharp wave-ripple (SWR).

**Figure supplement 2.** Sharp wave-ripple (SWR) properties are maintained regardless of mossy cell (MC) overlap.

each area of the Venn diagram is shown in *Figure 6—figure supplement 1*. These data suggest that the more MCs recorded simultaneously, the greater the total number of predictable SWRs. Indeed, the pooled plot of a total of 88,925 SWRs recorded from 23 slices shows that the percentage of predictable SWRs increased as the number of simultaneously recorded MCs increased (*Figure 6C*; p=1.00 × 10$^{-38}$, Jonckheere-Terpstra test). However, predictable SWR did not increase completely linearly with the number of MCs, but tended to increase sublinearly. Here, for the relationship between the number of MCs and the predictable SWR, we considered the sublinear increasing function as a general partial overlap problem and fit the data of *Figure 6C* to the exponential saturation function ($f$=1 – $\alpha^n$) by the least squares method. As a result, we obtained the function

$$p=1-0.920^n,$$

where $p$ and $n$ represent the ratio of predictable SWRs to the total SWRs and the number of MCs, respectively; the base of the exponential term $\alpha$ was 0.920 with a 99% confidence interval of [0.907, 0.933]. If the coverage of SWRs by MCs assumes random sampling of SWRs, the value of 1 minus $\alpha$ should be equal to the mean ratio of predictable SWRs by a single MC, which can be obtained from *Figure 4A*. Indeed, the value was 0.080 (=1–0.920), which was not statistically different from the mean ratio of predictable SWRs (0.092±0.048; Z=0.250, p=0.401, Z test). Therefore, according to this approximation function, roughly 8 MCs can predict 50% of the total SWRs, and 27 MCs can predict 90% of the SWRs. This sublinearity arises because there exist a small number of cases where multiple MCs predicted the same SWR (overlapping SWR). We calculated the percentage of overlapping SWRs that were predictable by both of any two MCs to the total SWRs for each recording dataset (*Figure 6D*; median = 1.56, SD = 2.04, $n$=113 MC pairs). The percentage of overlapping SWRs was sparse from record to record but never exceeded 10%, and this result confirms again that overlapping SWRs were in the minority. To characterize the overlapping SWRs, we analyzed their waveforms. The amplitudes of overlapping SWRs had a slight but significant positive correlation with the number of MCs that could predict the SWRs (*Figure 6E*, p=1.59 × 10$^{-5}$, Jonckheere-Terpstra test), suggesting that smaller SWRs are covered by a smaller number of MCs, while larger amplitude SWRs are covered

by a larger number of MCs. Other waveform properties, including the duration of SWRs, the peak oscillation frequency of ripples during SWRs, the fast Fourier transform power, and the area under the curve of SWRs, did not depend on the number of predicting MCs (*Figure 6—figure supplement 2*).

## Discussion

In the present study, we performed simultaneous in vitro whole-cell recordings of up to five hilar MCs in hippocampal slices and in vivo whole-cell recordings of single MCs from anesthetized mice and found that MCs responded to SWRs with their *Vm*s, which could predict the waveforms of SWRs in the CA3 region. The accuracy of predicting SWR waveforms improved with the number of simultaneously recorded MCs, but the *Vm*s of even a single MC could predict SWRs more accurately than expected by chance. Predictability was heterogeneous among SWRs, but a subset of SWRs predicted by one MC differed greatly from those predicted by another MC, and only a small fraction of SWRs was predictable by multiple MCs. As a result, the proportion of predictable SWRs increased efficiently with the number of simultaneously recorded MCs, and more than 30% of the total SWRs were predictable when five MCs were recorded.

In this study, we used a novel mathematical approach to correlate the upstream neural dynamics with the downstream neural response. In general, it is difficult to elucidate the relationships between the intracellular activity of neurons and the extracellular activity of neurons. In our preliminary studies, we performed several traditional experiments aimed at identifying a function that could correlate multiple *Vm*s with LFPs. However, conventional multivariate and regression analyses proved insufficient to describe LFPs in relation to *Vm*s. Specifically, although waveform prediction was possible using simple linear regression methods such as linear regression and ridge regression based on RMSE evaluation, it was not possible to reproduce the high-frequency band, which is an important feature of SWR. We therefore designed a neural network that was inspired by the autoencoder networks (*Hinton and Salakhutdinov, 2006*). The autoencoders have a bottleneck hidden layer that extracts low-dimensional codes in high-dimensional input vectors and are applicable to efficient data compression. Our neural network contains a bottleneck layer with 500 nodes, while its input length depends on the number of MCs. Using this simple neural network, we were able to predict LFP waveforms more accurately than chance prediction from the *Vm* dynamics of one to five MCs and were also able to reproduce the high-frequency band of the SWR. To evaluate the predictive performance of the model, we used RMSE as a key metric. We found that the RMSEs for predictions using real data were significantly lower than those obtained using shuffled data, demonstrating that MC *Vm*s contain structured, SWR-specific information. Had no difference been observed, it would suggest that the MC activity might only reflect nonspecific background input rather than meaningful information about individual SWRs. Similar results were also observed in the single MC analysis. We then focused on SWRs for which predictions based on real data had significantly lower RMSEs than those from shuffled data and defined these as 'predictable SWRs', i.e., SWR events whose information is encoded in the *Vm* dynamics of a given MC. Therefore, we quantified the proportion of SWRs that each MC could predict. Thus, we applied RMSE not only as a measure of prediction accuracy, but also as a quantitative proxy for the amount and diversity of SWR-related information embedded in the subthreshold activity of MCs.

In general, neural information diverges from the entorhinal cortex as it passes through the larger network formed by the granule cell layer and is subsequently compressed into the smaller network represented by the CA3 cell layer. This process inherently carries a risk of information loss. To counteract this, a backprojection mechanism via MCs has been proposed to prevent such loss (*Myers and Scharfman, 2011*). These theoretical models incorporating this mechanism show improved pattern separation and memory capacity, with results more consistent with experimental data compared to models lacking backprojection. However, it remained unclear what specific information individual MCs receive during backprojection. Our results show that CA3 SWR is distributed and encoded in the MC population, and that even though the number of MCs is smaller than that in other regions, it is possible to reproduce about 30% of the SWR in CA3 from the membrane potential of only five MCs. Based on these results, we propose that MCs not only help prevent information loss through backprojection, but also receive specific, experience-related memories that have been reactivated and temporally compressed within the CA3 region during SWRs. Therefore, information in backprojection is based on internally generated memory traces and is likely fundamentally different from

neural information transmitted via conventional feedforward pathways, such as the trisynaptic circuit (EC→DG→CA3→CA1), which primarily conveys self-motion and external environmental information as input from the neocortex via the entorhinal cortex (*Bicanski and Burgess, 2020*). Our recent study has shown that neurons in the medial entorhinal cortex can encode the future projected location of animals, forming a predictive grid representation that supports forward planning during spatial navigation (*Ouchi and Fujisawa, 2024*). In contrast to such forward-looking representations, the back-projection of SWRs is thought to reflect the reverse transmission of internally generated sequences, suggesting a complementary mode of information flow within the hippocampal-entorhinal system. However, it is important to note that the analysis in this study is based on slice experiments and subthreshold activity. The types of memory information that are backprojected and distributed to the MC, and how they differ from memory information transmitted via feedforward pathways, require verification through in vivo experiments. These are open questions that need to be addressed in future experiments in awake animals.

In this study, we performed MC patch-clamp recording both in vivo and in vitro and clarified that SWR can be predicted from *V*m of MC in both cases. However, there are three caveats to the interpretation of these data. First, the in vivo SWR cannot be exactly the same as the in vitro SWR; note that in vitro SWR has some similarities to in vivo SWR, such as spatial and spectral profiles and neural activity patterns (*Maier et al., 2009*; *Hájos et al., 2013*; *Pangalos et al., 2013*). The same concern applies to MC synaptic inputs. The in vivo *V*m data may contain more information compared to the in vitro single MC data, because the entire projections that target MCs are intact, resulting in a complete set of synaptic inputs related to SWR activity, as opposed to slices where connections are severed. While we recognize these differences, it is also very likely that there are common ways of expressing information. Second, since the in vivo LFP recordings were obtained from the CA1 region, it is possible that the CA1-SWR receives input from the CA2 region (*Oliva et al., 2016*) and the entorhinal cortex (*Yamamoto and Tonegawa, 2017*). In addition, urethane anesthesia has been observed to reduce subthreshold activity, spike synchronization, and SWR (*Yagishita et al., 2020*), making it difficult to achieve complete agreement with in vitro SWR recorded from the CA3 region. Finally, although we were able to record from MC *V*m during in vivo SWR in this study, we recorded from a single MC in our in vivo experiment, in contrast to the in vitro dataset where recordings from up to five MCs were obtained. To perform the same analysis as in the in vitro experiment, it would be desirable to record LFPs from the CA3 region and collect data from multiple MCs simultaneously, but this is technically very difficult. In this study, it was difficult to directly clarify the consistency between CA3 network activity and in vivo MC synaptic input, but the fact that the SWR waveform can be predicted from in vivo MC *V*m in CA1-SWR may be the result of some CA3 network activity being reflected in CA1-SWR. It is undeniable that more accurate predictions would have been possible if it had been possible to record LFP from the CA3 regions in vivo.

Our analysis focused on SWRs that could be predicted from individual MCs. The synaptic transmission to the MC is influenced by many factors, not only from the CA3 region, but also from other sources such as hilar interneurons, semilunar granule cells, and GCs (*Larimer and Strowbridge, 2008*; *Larimer and Strowbridge, 2010*; *Scharfman and Schwartzkroin, 1990*); note that it is known that there are projections from the CA3 region to inhibitory interneurons in the hilus (*Kneisler and Dingledine, 1995*). Conversely, in the case of MC, where there is much spontaneous activity, the fact that the SWR can be predicted from the shape of the EPSP at the time of SWR occurrence suggests that the information of the SWR is strongly reflected in the synaptic inputs of MC. However, an important consideration in interpreting this study is that the resting membrane potential of MCs is highly active even under normal conditions, often exhibiting barrage activity where EPSPs accumulate immediately after their generation (*Scharfman, 1993*). Barrages were occasionally observed in the MC EPSP used in this study during SWRs. Therefore, the possibility that these EPSPs included input from GCs cannot be ruled out. Thus, it should be noted that the MC EPSPs used for machine learning do not necessarily reflect input purely from CA3 SWR. Nevertheless, the robustness of decoding performance despite such confounds further supports the specificity of the CA3 SWR to MC backprojection. On average, each MC accounts for about 10% of the SWRs. This may be due to the fact that only a small number of MCs share the coding of SWRs. Surprisingly, we found that more than 30% of the SWRs were predictable by at least five MCs. On the other hand, since MCs are biologically vulnerable (*Scharfman, 2016*), it is important that multiple MCs simultaneously take care of the same SWR to avoid the risk of

information loss due to neuronal cell loss. We attribute the sublinear rather than linear increase in the percentage of predictable SWRs with increasing number of MCs to such overlapping MCs. Moreover, by applying the data to approximate functions, we found that about 8 MCs could predict 50% of the total SWR and 27 MCs could predict 90% of the SWR. Given that the hippocampal formation contains a total of 30,000 MCs, the number of MCs seems to be sufficient to cover the entire SWR information (*Amaral et al., 1990*; *Jinno and Kosaka, 2010*) and instead of the amount of information we expected, the MCs received more information from the CA3 region. Therefore, the MC layer is likely to have both redundancy and independence in encoding information. In addition, the finding that SWR is more predictive when the recorded location of the MC is near the lower blade of the DG was unexpected. It cannot be ruled out that preparing the acute slices may have severed specific synaptic connections due to tissue dissection, thereby influencing the results.

SWRs originating in the CA3 region are likely to be reflected in the MC as synaptic activity and transmitted to the DG (*Penttonen et al., 1997*; *Swaminathan et al., 2018*). Indeed, it has also been observed in in vitro experiments that some, but not all, MC spikes are likely to be recruited during SWR (*Swaminathan et al., 2018*); however, MCs are present in much smaller numbers than CA3 pyramidal cells and DG granule cells (*Jinno and Kosaka, 2010*). Thus, it is unclear how hippocampal information is efficiently encoded despite the limited capacity of the MC population. Previous studies have shown that SWRs are reflected as synaptic activity in MCs at single-cell level (*Ouchi et al., 2017*; *Swaminathan et al., 2018*). We observed distributed coding by MC populations by simultaneously recording SWRs in CA3 regions and up to five MCs. Furthermore, SWRs with smaller amplitudes were processed by a smaller number of MCs, whereas SWRs with larger amplitudes were covered by a larger number of MCs. In addition, we found that virtually all MCs responded reliably to each SWR, suggesting that SWR information is not sparsely encoded, but encoded by one or more MCs. Given that different SWRs may encode information that correlates with different experiences, it is also possible that the activity of individual MCs may play a role in encoding different experiences via SWRs. Indeed, several in vivo studies have confirmed that MC activity is involved in the space encoding (*Danielson et al., 2017*; *GoodSmith et al., 2017*; *Senzai and Buzsáki, 2017*; *Bui et al., 2018*; *Huang et al., 2024*). However, in vivo experiments have not extensively investigated the relationship between SWR and MC. The significance of the fact that the SWRs recorded from CA3 are distributed and encoded in MC populations is that it not only shows the transmission pathway from CA3 to MC, but also reveals the information below the threshold that leads to firing, and in a broad sense, it approaches the mechanism by which information processing by neuronal firing. The expression of synaptic input to the MC is not uniform, but varies in a variety of ways according to the pattern of SWR. Based on previous research showing that diversity is important for information representation (*Padmanabhan and Urban, 2010*; *Tripathy et al., 2013*), it is possible that this heterogeneity in membrane potential levels, rather than the all-or-none output of neuronal firing activity, is the key to encoding more precise information. In this respect, our research, which focuses on information encoding at the subthreshold level, may be able to extract even more information than information encoded by firing activity. This study not only provides important insights into the information processing of neural circuits at the bottleneck layer, but also introduces a new approach that employs machine learning to bridge different levels of time-series data, such as LFP and *Vm*.

## Methods
### Slice preparation

Acute slices were prepared from the medial to the ventral part of the hippocampus. On postnatal 21–29, male ICR mice were deeply anesthetized with isoflurane and decapitated. Their brains were rapidly removed and horizontally sliced (400 μm thick) at an angle of 12.7° to the fronto-occipital axis using a vibratome and an ice-cold oxygenated (95% $O_2$, 5% $CO_2$) cutting solution consisting of (in mM) 222.1 sucrose, 27 $NaHCO_3$, 1.4 $NaH_2PO_4$, 2.5 KCl, 1 $CaCl_2$, 7 $MgSO_4$, and 0.5 ascorbic acid. This cutting angle preserved more Schaffer collaterals in the slices and was suitable for the reproducible generation of SWRs (*Mizunuma et al., 2014*). Slices were incubated at 37°C for 1.0 hr and maintained at room temperature for at least 30 min in a submerged chamber filled with oxygenated artificial cerebrospinal fluid (aCSF) containing (in mM) 127 NaCl, 3.5 KCl, 1.24 $NaH_2PO_4$, 1.3 $MgSO_4$, 2.4 $CaCl_2$, 26 $NaHCO_3$, and 10 D-glucose.

## In vitro multiple patch-clamp recording

Recordings were performed in a submerged chamber perfused with oxygenated aCSF at a rate of 3–5 ml/min and a temperature of 33–35°C. Whole-cell current-clamp recordings were simultaneously obtained from up to five MCs in the dentate hilus, which was visually targeted using infrared-differential interference contrast microscopy. Patch pipettes (3–13 MΩ) were filled with a potassium-based solution consisting of (in mM) 135 potassium gluconate, 4 KCl, 10 HEPES, 10 creatine phosphate, 4 Mg-ATP, 0.3 Na$_2$-GTP, 0.3 EGTA, 5 QX-314, and 0.2% biocytin. LFPs were recorded from the CA3c stratum pyramidale using borosilicate glass pipettes (0.5–1.2 MΩ) filled with aCSF. The signals were digitized at a sampling rate of 20 kHz. The data criterion for series resistance was less than 45 MΩ, and the change rate of series resistance before and after the recording was <30%. The data were adopted when the mean resting membrane potential was <–50 mV and when $Z$-scores of the mean membrane potentials for 30 s were between –2 and 2. Data were included in the analysis if the recorded cells had a membrane capacitance higher than 45 pF (*Hedrick et al., 2017*) and had spines and thorny excrescences on their proximal dendrites (*Murakawa and Kosaka, 2001*; *Amaral, 1978*; *Frotscher et al., 1991*), which are electrophysiological and morphological markers of MCs. Since electrophysiological characteristics may include other cell types, we first confirmed the presence of thorny excrescences in the proximal dendrites using biocytin filling during recording. Only cells that met both criteria were classified as MCs and included in the analysis. The CA3 recording site was always located in the CA3 pyramidal cell layer closer to the hilus, which is the inner part of the CA3 pyramidal cell layer, assuming that a straight line connects the suprapyramidal edge and the infrapyramidal edge.

## In vivo patch-clamp recording

Male ICR mice were used in the experiments on postnatal days 28–40. Before the surgery, the mice were exposed to an enriched environment for 30 min to increase the occurrence of SWR during the experiment (*Okada et al., 2024*). The mice were anesthetized with urethane (1.95 g/kg, i.p.), and 1.0% lidocaine was subcutaneously applied to the surgical region. The anesthetized mice were fixed with a metal head-holding plate. The surgical procedures were described previously in detail (*Matsumoto et al., 2016*). Current-clamped recordings were obtained from MCs at depths of 940–1150 μm from the dorsal alveus using borosilicate glass pipettes (3–8 MΩ). Patch pipettes were filled with a potassium-based solution consisting of (in mM) 135 potassium gluconate, 4 KCl, 10 HEPES, 10 creatine phosphate, 4 Mg-ATP, 0.3 Na$_2$-GTP, 0.3 EGTA, and 0.2% biocytin. Once a satisfactory whole-cell recording was obtained, the firing properties of the neuron were identified by applying a 500 ms current from –100 to +100 pA in increments of 20 pA. Cells were discarded when either the mean membrane potentials exceeded –40 mV or the firing patterns were fast-spiking. Tungsten electrodes (FHC, USA) were used for extracellular recordings from the striatum oriens to the radiatum of hippocampal CA1 subregion. The LFP signals were amplified using DAM80 AC differential amplifiers (World Precision Instruments). All signals were sampled at 20 kHz using a Digidata 1440A (Molecular Devices). Tungsten electrodes were labeled with 1,1'-dioctadecyl-3,3,3',3'-tetramethylindocarbocyanine (DiI, Invitrogen) which was dissolved in acetone and methanol (1:1) at a concentration of 40 mg/ml. Their tracks were visualized post hoc. Data were adopted when the electrode tips were placed within the CA1 region.

## Histology

For the visualization of patch-clamped neurons from the in vitro experiment, the slices were fixed in 4% paraformaldehyde and 0.05% glutaraldehyde at 4°C for more than 24 hr. After the solution was washed out, the sections were incubated with 0.2% Triton X-100, a streptavidin-Alexa Fluor 594 conjugate (1:500) and 0.4% NeuroTrace 435/455 Blue Fluorescent Nissl stain (Thermo Fisher Scientific; N21479) overnight at 4°C. The tissue sections from the in vivo experiment were incubated with a rabbit primary antibody for GluR2/3 (Merck Millipore; AB1506; 1:200) overnight and then with a secondary goat anti-rabbit IgG (Cell Signaling Technology; #4412; 1:500) for 6 hr. Fluorescent images were acquired using a confocal microscope (FV1200, Olympus or BX61WI, Olympus) and subsequently merged.

## SWR/ΔVms detection

LFP traces for the in vitro experiments were bandpass filtered at 2–30 Hz, and the SWR peak times were determined at a threshold above the mean + 5 × SD of the baseline noise (*Mizunuma et al., 2014*; *Norimoto et al., 2018*). SWR onset times were determined at a threshold above the mean + SD of the baseline noise in up to 50 ms before the SWR peak time. On the other hand, LFP traces for the in vivo experiments were bandpass filtered at 120–249 Hz, and the SWR peak times were determined at a maximum value above the mean + 3 × SD of the baseline noise. The detected events were scrutinized by eye and manually rejected if they were erroneously detected. For the amplitude of an SWR-triggered fluctuation in the subthreshold membrane potentials of MCs (ΔVms), the mean Vm (baseline) was determined 20 ms before the peak time of each SWR and subtracted from the maximal Vm within –30 to +40 ms from the SWR peak time.

## Prediction of SWR waveforms from Vms and the model architecture

A neural network model was designed to predict the SWR waveforms from one to five simultaneously recorded Vms using Python 3.8, TensorFlow 2.3.1, and Keras 2.4.3. The GPU specification applied was NVIDIA Quadro RTX 4000. The architecture of the model is shown in *Figure 3A*. Our model has an encoder-decoder structure. The encoder compresses the input (Vms that correspond to SWR) to a lower-dimensional representation and extracts features from the input, whereas the decoder reconstructs the final output (SWR waveforms) from the compressed vector. Before training, the traces of the Vms and SWR waveforms were pre-processed by scaling them between 0 and 1. In the encoding operation for in vitro data, each Vm trace for ±50 ms relative to the SWR peak time (size 2001) was initially concatenated and passed through a fully connected layer. In this operation, the input was compressed to a feature matrix of reduced size (size 500) (*Figure 3A*). On the other hand, for in vivo data, each Vm trace for ±100 ms relative to the SWR peak time (size 4001) was initially concatenated and passed through a fully connected layer. In this operation, the input was compressed to a feature matrix of reduced size (size 100) (*Figure 3—figure supplement 3*). Our model was implemented using the Python deep learning library Keras and the TensorFlow backend. The activation function was set to *ReLU* when compressing from encoder, and the activation function was set to *sigmoid* when restoring from latent variables. The network was optimized by adaptive moment estimation (Adam) with a learning rate of 0.001. The parameters for the optimizer Adam were as follows: $\beta 1$ (an exponential decay rate for the first moment estimates)=0.9, $\beta 2$ (an exponential decay rate for the second moment estimates)=0.999, and $\varepsilon=10^{-7}$; the default values were used for the other parameters. Our model was trained to produce the original SWR waveforms from Vms at the corresponding time points. To increase the number of datasets, we divided the quintuple, quadruple, and triple patch-clamp recording datasets into subsets with fewer MCs. Therefore, the analyzed data included predictions of the same SWR waveform with different MC combinations. To assess the model performance on the entire dataset, fivefold cross-validations were used. Each dataset was equally divided into 5 subsets in order of occurrence so that all SWRs would be test data at least once; 1 subset was used as the test data, while the remaining 4 subsets were further divided into 10 subsets (1 subset was used as the validation data, and the remaining 9 subsets were used as the training data). If the number of SWRs is not divisible by 5, to keep the number of SWRs used in the test data constant, some of the same SWRs are used as test data in the fourth and fifth rounds of cross-validation. For each combination of inputs (i.e. one Vm, two Vms, three Vms, four Vms, and five Vms), the training lasted 100 epochs with a batch size of 16, and the RMSEs were calculated to assess how well the model predicted new data that were not used for training. The following was performed as a pre-processing step to calculate RMSEs. The predicted SWR waveform and the original SWR waveform corresponding to 10–20 ms were extracted, averaged, and then the difference between them was calculated and used as the correction value. After subtracting this correction value from the predicted SWR, the RMSE was calculated. As a randomized control, surrogate data were produced by shuffling the combinations of Vms; i.e., the order labels of SWRs were exchanged within each cell. For each dataset, 100 surrogates were produced, the model was also trained using these shuffled data, and the RMSE was calculated.

## MDS space

To ensure that similar SWRs are placed close together and to visualize them in two dimensions, the MDS algorithm was used in this study. The plot reduced to two dimensions is referred to as MDS

space. The variable used in the MDS algorithm is the matrix of RMSEs (i.e. dissimilarities) between simultaneously recorded SWRs.

## Semicircular diagrams of the DG

Data analyses were performed using ImageJ. The anatomical shape of the DG was normalized as a semicircular diagram based on Nissl staining in order to visualize the distribution of recorded MCs. The subgranular zone was defined as the innermost layer of the granule cell layer where cells are densely distributed. The location of each MC was determined by its angle and distance in the DG as follows. For the definition of the angle of each MC, the edge of the CA3 pyramidal cell layer was defined as the center of the hilus, a line connecting the hilus center with the suprapyramidal blade (i.e. the area closest to CA1) edge of the granule cell layer was defined as 0°, and the infrapyramidal edge was defined as 180°. For the distance of each MC in the hilus, the total distance from the edge of the CA3 pyramidal cell layer to the subgranular zone was determined, and the relative position of the MC was normalized to this distance. Using these methods, the angle and distance were determined for all MCs. Detailed information on this method is provided in the previous publication (*Koyama et al., 2012*).

## Spatial entropy

Spatial entropy was calculated to investigate whether the distribution of recorded MCs in the semi-circular diagram is related to the predictability of SWR (*Figure 4B and C*). The spatial entropy for the semicircular diagram was calculated by converting the polar coordinate values to Cartesian coordinate values and calculating the Euclidean distance between each two points. This resulted in an average of 5% predictable SWRs for each MC and its four nearest neighbors in the semicircular space. This proce-dure was repeated to collect averages for all MCs. The spatial bias of the average % predictable SWR was then quantified by spatial entropy. Spatial entropy was calculated as $-\Sigma P_i \log_2 P_i$, where $P_i$ is the probability frequency of the mean % predictable SWR of the MC with MC identification number $i$. The chance level of spatial entropy was estimated by 10,000 surrogates in which the value of % predictable SWR was permuted across the MCs. When the spatial entropy is significantly lower than the shuffle, the MCs that are more likely to predict SWR indicate a bias in anatomical location.

Next, to determine if there is a bias in the SWR that each MC can predict, the spatial entropy was calculated (*Figure 5A and B*). For each spatial entropy predicted by a single MC, the prediction rate was computed from $RMSE_{real}$ and $RMSE_{surrogate}$, where $RMSE_{real}$ and $RMSE_{surrogate}$ represent the RMSEs between the original SWR waveform and the SWR waveforms predicted by neural networks trained using real $V$ms and surrogate $V$ms, respectively. SWR was determined to be significantly predictive when $RMSE_{real}$ was significantly lower than $RMSE_{surrogate}$ calculated 100 times. For each SWR and its four nearest neighbors in the MDS space, their five binarized predictions were averaged. The calculation was replaced by 1 if the SWR was determined to be significantly predictive and 0.1 for otherwise. This procedure was repeated, and the averages were collected for all SWRs. Then, the spatial bias of the mean prediction scores was quantified by spatial entropy, which was calculated as $-\Sigma P_i \log_2 P_i$, where $P_i$ is the probability frequency of the mean binarized prediction of MCs with the SWR identity number $i$. SWR identity number is defined according to the order of SWRs occurring within the dataset. The chance level of spatial entropy was estimated by 1,000,000 surrogates in which SWRs were permuted within the MCs. If the spatial entropy is significantly lower than the shuffle, we interpret this as a bias in the SWR that can be predicted by the MC.

## Data acquisition and analysis

Data were analyzed offline using custom-made MATLAB routines (MathWorks, Natick, MA, USA), and the summarized data are reported as the means ± standard deviations (SDs) unless otherwise noted. p<0.05 was considered statistically significant. Since the data did not follow a normal distribution, a nonparametric test was performed.

## Acknowledgements

We thank S Fujisawa for insightful comments. This work was supported by JST ERATO (YI: JPMJER1801), the Institute of AI and Beyond of the University of Tokyo (YI), AMED-CREST (YI: 24wm0625401h0001, 24wm0625502s0501, 24wm0625207s0101, 24gm1510002s0104), JSPS Grants-in-Aid for Scientific

Research (YI: 18H05525, 22K21353), Grant-in-Aid for JSPS Fellows (AO: 18J14574, 20J01255), Grant-in-Aid for Early-Career Scientists (AO: 23K14231; NM: 25K18705), RIKEN Special Postdoctoral Researchers Program research grant (AO: 202401061042), and the JST CREST program (TT: JPMJCR23N2).

## Additional information

### Funding

| Funder | Grant reference number | Author |
|---|---|---|
| Japan Science and Technology Agency | 10.52926/jpmjer1801 | Yuji Ikegaya |
| The Institute of AI and Beyond of the University of Tokyo | | Yuji Ikegaya |
| AMED-CREST | 24wm0625401h0001 | Yuji Ikegaya |
| AMED-CREST | 24wm0625502s0501 | Yuji Ikegaya |
| AMED-CREST | 24wm0625207s0101 | Yuji Ikegaya |
| AMED-CREST | 24gm1510002s0104 | Yuji Ikegaya |
| Japan Science and Technology Agency | 18H05525 | Yuji Ikegaya |
| Japan Science and Technology Agency | 22K21353 | Yuji Ikegaya |
| Japan Science and Technology Agency | 18J14574 | Ayako Ouchi |
| Japan Science and Technology Agency | 20J01255 | Ayako Ouchi |
| Japan Science and Technology Agency | 23K14231 | Ayako Ouchi |
| Japan Science and Technology Agency | 25K18705 | Nobuyoshi Matsumoto |
| RIKEN | 202401061042 | Ayako Ouchi |
| Japan Society for the Promotion of Science | 10.52926/jpmjcr23n2 | Taro Toyoizumi |

The funders had no role in study design, data collection and interpretation, or the decision to submit the work for publication.

### Author contributions

Ayako Ouchi, Conceptualization, Data curation, Software, Formal analysis, Supervision, Funding acquisition, Validation, Investigation, Visualization, Methodology, Writing – original draft, Project administration, Writing – review and editing; Taro Toyoizumi, Funding acquisition, Validation, Methodology; Nobuyoshi Matsumoto, Software; Yuji Ikegaya, Conceptualization, Resources, Funding acquisition, Methodology, Writing – original draft, Writing – review and editing

### Author ORCIDs

Ayako Ouchi https://orcid.org/0009-0001-9285-1763
Taro Toyoizumi https://orcid.org/0000-0001-5444-8829
Nobuyoshi Matsumoto https://orcid.org/0000-0002-0426-9416
Yuji Ikegaya https://orcid.org/0000-0003-2260-8191

### Ethics

Animal experiments were approved by the Animal Experiment Ethics Committee of the University of Tokyo (approval number: P29-9) and performed in accordance with the University of Tokyo guidelines

for the care and use of laboratory animals. These experimental protocols were carried out in accordance with the Fundamental Guidelines for Proper Conduct of Animal Experiment and Related Activities in Academic Research Institutions (Ministry of Education, Culture, Sports, Science and Technology, notice no. 71 of 2006), the Standards for Breeding and Housing of and Pain Alleviation for Experimental Animals (Ministry of the Environment, notice no. 88 of 2006) and the Guidelines on the Method of Animal Disposal (Prime Minister's Office, notice no. 40 of 1995). All animals were housed under a 12 hr dark-light cycle (light from 07:00 to 19:00) at 22 ± 1°C and had ad libitum access to food and water.

### Decision letter and Author response
Decision letter https://doi.org/10.7554/eLife.97270.sa1
Author response https://doi.org/10.7554/eLife.97270.sa2

## Additional files

### Supplementary files
Supplementary file 1. Recording time and number of sharp wave-ripple (SWR) events in each slice. Some data were recorded multiple times from a single slice, and the quality of the recordings was confirmed by checking the access resistance and intrinsic electrophysiological properties of the mossy cells (MCs) did not change significantly between recordings. When there were multiple recordings, each value was entered in the same cell. The data thus obtained were concatenated in machine learning and treated as a single dataset. Note that 190801_2 and 190927_1 have overlapping slice IDs and different cell numbers. In this case, the MCs to be recorded are common, but the recording times (i.e. SWR events) used for the actual analysis do not overlap.

MDAR checklist

### Data availability
All data generated or analyzed during this study are included in the manuscript. Original datasets and codes are provided on GitHub (copy archived at *Ouchi, 2025*).

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
