## [Editor Report]

The study shows that activity of dentate gyrus mossy cells encode information from sharp wave-ripple complexes (SWRs) from the adjacent CA3 region. The study used difficult methods such as recording from multiple mossy cells simultaneously, as well as deep learning, which is impressive. The findings are fundamental in significance because they show a relationship between mossy cells and sharp wave ripples that has not been appreciated before, and the strength of evidence is compelling.

---

## [Decision Letter]

**Decision letter after peer review:**

Thank you for submitting your article "Distributed encoding of hippocampal information in mossy cells" for consideration by *eLife*. Your article has been reviewed by 3 peer reviewers, and the evaluation has been overseen by a Reviewing Editor and Laura Colgin as the Senior Editor.

Essential Revisions

1) Address the conceptual questions of Reviewers, especially Reviewer 1.

2) Clarify the deep learning approach as noted by all Reviewers.

3) Consider the statistical questions and revise as appropriate as noted by Reviewer 3.

*Reviewer #1 (Recommendations for the authors):*

Detailed additional concerns.

A ref is needed for CA3 neurons send input to MCs. I would suggest PMID: 7884451; PMID: 9157312 Note CA3 also sends input to hilar and GCL interneurons. For that I would cite PMID: 8008190, PMID 7473243, PMID: 8300905, PMID: 2358523.

Line 62

"there is no direct axonal projection from CA3 pyramidal cells to DG granule cells Fujise and Kosaka, 1999"

This is not likely to be true. Li et al. (PMID above) showed CA3 axons in the GCL and IML. However, I do not believe synapses were proven. Still, it is highly suggestive. Buckmaster showed similar findings in primate. PMID: 11135261

Line 63

"An estimated 15,000 MCs reside in the rat hippocampal formation, whereas approximately 300,000 and 2,450,000 CA3 pyramidal and granule cells exist, respectively"

I am not sure these numbers are correct. As a start, see PMID: 17765743, PMID: 17475251 Perhaps the authors are referring to a particular species and if so that should be discussed.. Finally, estimating GC numbers is very difficult because they overlap even in thin sections. So this needs to be considered carefully. See PMID: 29311853, PMID: 1793176, PMID: 3292009

Since it is not clear what the numbers are, it might be useful to show a range, and implement the deep learning with several possible values. If the results are the same that would make the results more convincing.

Line 66. Same caution needs to be made here

Line 103

I think the authors mean cells were excluded if capacitance was <45pF (please explain the reason) and they were included if they had spines and thorns.

Refs are needed that these are characteristics of MCs.

Line 109

Consider excluding cells with such instability that they were not held >1 min.

How were others designated healthy?

Line 465

Exposing mice to an enriched environment means what?

Why was this done? What were the effects? Was there a control?

Line 479

Why was -40 a cut off? That seems very depolarized and unhealthy.

How were MCs distinguished from CA3 or GCs? What about slow spiking hilar interneurons?

What were the brief inward currents exactly?

Line 480

What was the tungsten electrode exactly? What part of CA1 was recorded? Why? To record SW? If SW were recorded in CA1, the layer should be consistent. Was it checked?

Exactly how was DiI used?

Why were SW recorded in CA3 in vitro and CA1 in vivo?

Was the site in CA3 in vitro consistent across slices? How was it positioned so it was consistent?

Why is the term SW used, not SWR?

Line 503

'The detected events were scrutinized by eye and manually rejected if they were erroneously detected “

Please explain.

Line 500

Were definitions of SW similar in vivo and in vitro? Were the conclusions from SW in vitro confirmed by SWs in vivo? The data seem mainly from slices.

Please explain deep learning in more detail. One has to read the results to infer how this was done and many other aspects of the methods. The figure legends are not very helpful in this regard, making it hard to read the paper.

Please explain in the methods what the authors mean by:

spatial entropy

RMSE

MDS space

binarized prediction

significantly predictive

SW identity number

chance level of spatial entropy

SW permeation by MC

Line 562

How was the subgranular zone defined? Consider a supplemental figure showing the method.

If slices were transverse, the DG is not a semicircle. If horizontal sections were made, it is not either. Therefore explain how the DG was simplified to a semicircle.

How was the edge of CA3 defined without a stain?

Statistics

Please add means and SEMs where n is the number of mice. This could be done in supplemental material.

Stats are not explained at all. This is a serious omission. For example, what were the parametric and non parametric tests, and what were the tests of normality to make these decisions? Or was it assumed all data were normal? Were ANOVA's two-way and if so were there interactions?

Figure 1

SW are not clearly SW. The traces are too compressed to tell. What are the criteria in vitro?

What are the criteria in vivo? If animals do not have immobility but are forced to be immobile, what relationship are the SWs of slices and anesthetized animals to SW in vivo in awake behaving animals?

For the expanded traces in Figure 1B what is the voltage calibration?

The traces need to be decompressed to tell when during the SW the MC activity starts. That is important so that one can tell if it was possibly monosynaptic or polysynaptic. However, it is recognized that with field potentials the onset of CA3 activity will be hard to know precisely. Therefore monosynaptic CA3-MC connections will be hard to confirm. If polysynaptic circuits are involved this is very important because it might be CA3 only activates MCs when secondary pathways become involved. Importantly, polysynaptic transmission is always variable compared to monosynaptic transmission. Together there are many reasons MC responses to SWs are variable as discussed above.

If MCs have different resting potentials in Figure 1B it would seem logical and not very informative to point out that their depolarizations in response to a SW are variable. An analysis of variability as it relates to resting potential could help. One reason is at depolarized potentials NMDA receptors will become important and broaden EPSPs of MCs.

If it is variable in the same MC at the same resting potential, that is nice but it is (1) not clear how much this has been observed (2) if that is surprising since SWs themselves are variable.

Figure 1 and elsewhere. Please add number of mice throughout.

I can't see traces in C. Here and elsewhere they are too small. Please explain MDS1 and MDS2. Furthermore, what are the traces from? What does the sphere of dots signify?

D. Based on the comments above, consider a plot of depolarizations that Is just for cells at a small range of membrane potentials to see if there is more consistency. Also consider other factors: latency from the onset of the SW, duration, area under the curve. Are these just as a variable or not? If in variable it would be insightful. Can variability be reduced by use of APV to block NMDA receptors, cutting away CA3a so only CA3b and c are present, or just c, etc? One could also block GC transmission with DCG-IV and GABAergic transmission to reduce polysynaptic influences.

In D, what is Vm# on the Y axis? Was this just done for the one slice with 5 MCs? What about other data? Was it reproduced? What was the finding here? Just that there is variability?

E. Why z standardize? What were the weak correlations (I just see one R value only) and were they statistically significant? what was the test?

Why are data from MCs pooled? In other words, why equate the 87 cells if they were from 23 slices and even fewer mice?

Here and elsewhere there are numerous impressive recordings from more than 1 MC. However, how are the relationships interpreted in MCs of the same preparation if sometimes there are 1, 2, and other times up to 5? One wonders because totals are often cited as if everything is pooled. Moreover, there is a correlation between the number of cells recorded in the same preparation and some outcomes. Because of that finding, I don't see that pooling is justified.

Figure 2. In B there is no GluR2/3 stain in the inner molecular layer where MC axons are located.

Why was 100-250 Hz used as a filter and what was the type of filter? Presumably this is to study ripples. But ripples are not typically considered over 200 Hz and below 100 Hz is often included. Also the figure shows ripples besides the two SW events- what were these- small SWs?

The inset needs a voltage calibration.

Figure 3. In A, the neural network is not explained thoroughly. What were the assumptions based on? What does a full connected link mean relative to one that is not connected? Why is there only unidirectional flow in light of the complex circuitry? What is the take home message in B, F? In C, what were the statistics? Please make symbols in D large enough to see. Explain the box and whisker plots. Were these data from one slice? Were they reproduced?

Please explain E.

Figure 4 Please explain the title. In A, what is the count of? In C, please explain the method of analysis of spatial entropy. The methods are not clear and the relationship of what is said in the methods about entropy to Figure 4C is not clear.

It seems here that it does not matter where the MC is located. Is that the message?

In D, E how was the single SW selected?

In Figure 5, What is "predicted deflection"? "dimensionally reduced"? What is the threshold for defining a SW event was predictable in A?

B. What is the spatial bias of a rate? Change distribution? Are "surrogates' the predicted events and non-surrogates are the actual SWs? What is the bottom line of this figure?

The very intriguing hypothesis in the abstract – i.e., a particular MC showed a more robust association with a particular SWR cluster, and the SWR cluster associated with one MC rarely overlapped with the SWR clusters associated with other MCs. – where is the evidence supporting this hypothesis? It seems only based on the complex analysis where there is a model driving the conclusions and the model may not be a good model.

Line 152. The authors find that the prediction error of real data was lower than shuffled data. Then they conclude from this that CA3 "information during SWs was at least partially preserved in the Vm responses of MCs" I see this as a major leap. All I see is that the deep learning had less of a prediction error for real data than shuffled data. Why is there an inference that this is meaningful in the way the authors state?

Line 160. The MCs were divided into subsets. Please explain what was done exactly. Why is it surprising that the more MCs are recorded simultaneously the better the prediction can be?

Line 169. Here the frequency range is 120-250 but in the figure it is 100-250.

Line 172. Please explain why the data suggest MCs "retain more information about SWRs"

Line 182. The SW waveform was predicted by the MCs but I am not sure that is interesting because the traces show that SWs are so similar from one example to the next. So it would be easy to predict.

Line 205. Only two intrinsic characteristics of MCs were examined so it seems hard to conclude that characteristics of MCs do not matter.

The fact that MCs in the area of the lower blade were more predictive could simply be due to greater connectivity with CA3 in that location. It would be good to investigate.

Line 211 What is meant by a dimensionality of SWs and reducing a SW to two dimensions?

The paragraph starting on page 224 is critical to the conclusion that is novel in the paper but it is not well explained. Therefore it is hard to be convinced. For example, is it justified to binarize prediction scores into 1 and 0? Is it reasonable to pool data from all MCs whether they were recorded with 1 other or 2 or more?

*Reviewer #2 (Recommendations for the authors):*

(1) Concerning more detailed model description

a – Details on model architecture: The manuscript briefly mentions the use of an encoder-decoder architecture with dense layers, but the rationale for choosing this architecture or the specific configuration of the layers used is not fully explained. The inclusion of such details, along with the rationale for the choice of activation functions (relu and sigmoid), would provide deeper insights into the model design.

b – Details on the training process: The paper would benefit from a more thorough explanation of the preprocessing steps applied to the data prior to training, as well as the rationale for the chosen loss function (root mean squared error, RMSE) and the detailed settings of the optimizer (e.g. learning rate, β values). This information is crucial for anyone attempting to replicate the study or apply the methodology to similar problems.

c – Details on data handling and evaluation metrics: Although cross-validation is mentioned, a clearer explanation of how the data sets were split and the criteria used for this split would be helpful.

d – Details on implementation and computational resources: Specifying the versions of TensorFlow, Keras and other dependencies used would help in replicating the computational environment. In addition, information about the required computational resources, such as GPU specifications (if applied), would provide practical insights into the feasibility of reproducing the study.

(2) Manuscript organization

The figures are effective in presenting the data, but the narrative in the text can be difficult to follow in places. Reorganizing the manuscript could facilitate better comprehension for readers. Specifically:

- Methodological descriptions should be confined to the Methods section.

- Theoretical and analytical concepts could benefit from succinct explanations where

they first appear.

- Transitions within the narrative should be smooth and well-motivated. For instance, the spatial analysis on page 7 requires a abrief introduction to provide context.

- A noteworthy result regarding the predictive value of MCs in relation to SWR information, currently discussed on page 11 (lines 300ff), could be more appropriately placed in the Results section.

(3) References

a – Hyde and Strowbridge (Nat Neurosci, 2012) have previously published research relevant to the topic; they used artificial stimuli to demonstrate encoding of temporal sequences into the MC network, suggesting mechanisms for short-term memory. The authors should acknowledge this study and discuss these earlier results in the context of their own findings.

b – Nitzan et al. (2020) undoubtedly serves as a valid reference for supporting the propagation of ripples from CA3/CA1 to cortical targets. However, since this phenomenon has previously been demonstrated and discussed, it would be preferable to cite earlier studies that originally identified or thoroughly investigated this propagation.

(4) Further comments

a – To maintain consistency in the literature, it is recommended that the authors align with established nomenclature. Specifically, 'SW' typically denotes slow waves, while 'SPW,' introduced by Buzsáki (1986) for sharp waves in the hippocampus, and SPW-R (or SWR), for sharp wave-ripple complexes, are the accepted terms. Using these conventions would enhance clarity and ensure alignment with the field's standard terminology.

b – "Input" versus "Response": The manuscript's language at times implies an active role of mossy cells. However, the analysis primarily concerns *synaptic inputs* from CA3 to MCs, indicative of upstream network dynamics, rather than direct responses (that is, spiking activity) of MCs. Since responses of MCs in the true sense were not the focus of this study, I recommend a thorough review and revision of the manuscript to accurately reflect this distinction.

c -Line 222f: "… a MC tended to predict a specific subset of SWs …" – A more appropriate (precise) formulation would imply that it is not MCs that predict a specific subset of SPWs, but the synaptic input that an MC receives from the upstream network.

d – Line 293: "Only a single MC was recorded in our in vivo study." – This sentence seems to be misleading: If I am not mistaken, the authors meant to write that their in vivo dataset consists of single MC recordings, as opposed to their in vitro dataset, which contains recordings of up to five MCs recorded simultaneously. Please rephrase this sentence.

e -Line 296: "…because the nerve fibers are preserved without sectioning in the in vivo preparations." – This statement, while not incorrect, is somewhat unspecific; the authors should perhaps specify that in vivo, CA3 projections targeting MCs are intact, resulting in a complete set of synaptic inputs related to CA3 network activity, as opposed to slices where connections are severed.

f -Line 313: "…the exponent α was 0.920 with a 99% confidence interval of [0.907,0.933]." – Please check the mathematical terminology used in the description of your equation. In the context of the given equation p=1−α^n, α serves as the base of the exponential term, and n is the exponent. Please correct.

g -Line 633: "… (in vivo) n = 273 SWs from 2 quintets." – Single MCs were recorded in vivo. Please correct.

h -Figure 2: The inset in panel B does not convincingly show the thorny excrescences. Can the authors add a microscopic image that shows this crucial feature of MCs more clearly?

i -Figure 3: Scale bars should be added to the traces displayed in panel B and F.

j -Figure S3: "Reproducibility of high frequency (120-250 Hz) increases…" – A more precise title would be "Reproducibility of 120-250 Hz coherence increases…"

k -Figure S4: The source of the predicted sharp wave/ripples (SWRs) is indicated as being from the CA3 region, yet the validation of prediction errors appears to be based on data from the CA1 region (main Figure 3). Please review and correct, if applicable.

*Reviewer #3 (Recommendations for the authors):*

1) Lines 69, 324: SWs do not propagate from CA3 to DG (other than volume conduction). Rather, the SW is a reflection in the LFP of local activity at the cellular level (spikes and synaptic inputs). It is the action potentials from CA3 cells that propagate the signal to drive mossy cells. I know what the authors mean, but it is important to be precise in one's language.

2) Line 72: SWs occur "primarily" during sleep and quiet wakefulness, but they also can occur during behavior, albeit much less frequently.

3) Line 73: "SWs represent a mechanism for reactivating or consolidating…." Although evidence in favor of this idea continues to accumulate, I don't think it is proven yet, and the words "are thought to" should appear before "represent".

4) Line 75: CA1 also outputs to the neocortex via direct projections back to deep layers of the EC

5) Line 80-81: The pioneering studies by Wilson & McNaughton (1994) and by Skaggs and McNaughton (1996) should be cited here.

6) Line 105: Do the authors mean "included" here? Why would you exclude cells with thorny excrescences on the dendrites?

7) Line 209: What does prediction "deflection" mean? This term is not defined.

8) Line 467: What was the purpose of exposing the mice to an enriched environment? I assume this was for a different study, but it should be clarified.

9) Figure S4 caption: The wording of this caption is confusing: "the SW of in vivo has a longer SW than that of in vitro"? Delete "that of"?

[Editors' note: further revisions were suggested prior to acceptance, as described below.]

Thank you for resubmitting your work entitled "Distributed encoding of hippocampal information in mossy cells" for further consideration by *eLife*. Your revised article has been evaluated by Laura Colgin (Senior Editor) and a Reviewing Editor.

The manuscript has been improved but there are some remaining issues that need to be addressed, as outlined below:

*Reviewer #1 (Recommendations for the authors):*

The authors showed in their responses that they considered the concerns and took time to try to address the concerns. The efforts are appreciated. The manuscript is improved.

However, several of the concerns were not well addressed.

One issue that may not have been clear is that an explanation of the methods to construct a model of the network and the methods to use the model to develop the conclusions are not clear. This is critical so that the readers understand how the data were used to lead to the conclusions. In parts of the response some information is provided and this needs to be added to the manuscript. In addition, the additional methods need to be provided.

Some other fundamental concerns are raised by what the authors mention in their response:

If MCs were recorded for 55-367sec, how could the authors know (1) the cell was healthy, and (2) the cell was a MC? Regarding (1), how did the authors assess the health and stability of the recording? It would seem they could not so they simply assumed it was healthy, despite only keeping the recording for a short time which one would think implies the cell was not healthy. That should be stated in the methods, one would think. Regarding (2), it is stated that cells were filled with biocytin, and one is shown. If no time was available to determine if the cell was a MC by physiology, then the biocytin is critical to identify that the cell was a MC. Was it the case that all cells were identified as MCs by biocytin, or was it only a subset? If some cells were not stained or not stained well enough, why were they included? There are many large cells in the hilus that are not MCs so just the size of the soma is not a good way to tell if a hilar cell is a MC.

If MC depolarizations could not be analyzed by AUC or duration because there were barrages, this would be consistent with hilar cell physiology shown previously. The problem is that if barrages occur, one does not know if a SPWR caused a depolarization very well. This is because a SPWR has a duration that is 50-200 msec and barrages of EPSPs during that time will make it seem like MC depolarizations were caused by a SPWR but they actually occurred independently. A great way to make this easier is to use DCG-IV to stop the majority of MC spontaneous activity because it stops the GC input to MCs. If the authors do not choose to do this, then they need to make a better case that SPWRs induce MC depolarizations.

I apologize for not having read the Impact statement earlier. The current impact statement is quite complex, and the main message is not clear. Please revise.

There are several assumptions that are not clearly based on data in the literature

1. CA3 SPW-Rs are transmitted to the granule cells (GCs) via MCs. CA3 projects to both MCs and hilar GABAergic neurons. Together they innervate GCs, as well as each other. Therefore SPW-Rs in GCs are not necessarily a product of CA3 excitation of MCs alone. This was mentioned before and the authors added references that were suggested and content, which is good. But they added it in different locations. The Introduction still reads as if CA3 innervates MCs and MCs excite GCs, producing GC SPWRs. This is an oversimplification. Also, note that if the algorithm does not take into account the circuitry correctly it could make it seem like MCs are responsible for the events they call GC SPW-Rs and MCs are not solely responsible.

2. Please note it also is not clear to this reviewer that LFPs like SPWRs occur in the GC layer. What is it that the authors think are "SPWRs" in recordings of the GC layer? Depolarizations? Dentate spikes?

3. It is confusing if the term SWR is referring to the ripple or the SPW or both. This is because SWR is defined as a ripple 80-200 Hz in one location of the paper. Furthermore, if it is, how would MCs produce a ripple in GCs if GABAergic neurons are not invoked? This seems to be a major point of confusion.

4. What is the reason why "MC ensembles" are assumed? It seems there is no basis for this. MCs are not interconnected, or are not interconnected much. Perhaps the term "ensembles" is not what the authors mean?

On line 74-76, the references that are cited did not show that SPWRs travel from CA3 to the DG by MCs. Pentonnen et al. did not prove MCs were critical. It was merely suggested. Scharfman did not either.

Please explain the reasoning in the Introduction more clearly. For example, when are the authors discussing CA3 neurons and when are they discussion other neurons. When are they discussing the ripple and when are they discussing the sharp wave?

Please explain the meaning of overcapacity on line 424. Why do the authors state that MCs form a layer? They are heterogeneously distributed. Throughout the Discussion it seems that the authors are making statements of fact based on their network model and this is not appropriate. In the model, for example, if the make MCs a layer, that does not mean they are in vivo. Please specify in the Discussion that the results are what the neural network found, not the data (especially lines 418 and following)

Lines 434-437 are not clear. What the authors say they show is that CA3 has robust excitatory input to MCs, but this has been shown. Furthermore the authors seem to say they show MC activity occurs during SPWRs, but that has also been shown. On the other hand, one study did not find MC activity during SPWRs were not so consistent (Swaminathan et al.). The differences from the current study are not discussed.

Other points:

The response to the first comment is not very strong. The argument that the study proved connections between CA3 and MCs that have not been shown before is incorrect. Perhaps the authors are saying something else?

In response to the second point the authors say their study shows the extent to which SWRs are encoded in MCs. I do not see the authors showed this. That conclusion is based on a very simplified model and methods that they do not describe well.

Explanations in the response are not always added to the paper. Please do so.

*Reviewer #3 (Recommendations for the authors):*

The authors have addressed most of my points, but one of the major points is still unresolved.

Original comment 4: Although a mixed-effects model is preferable, the authors' new analyses are sufficient to show the same point, and this comment is considered resolved.

Original comment 5: The author's response is unsatisfactory. They acknowledge that the problem I raised was correct, but they continue to show the result of this flawed statistical analysis in Figure 3C. Instead, they add a new supplementary figure with a paired t test (the rebuttal letter and the main text (line 179) state that this is Figure 3-supp Figure 5, but I believe it is really Figure 3-supp Figure 1, as supp Figure 5 does not exist). They need to remove Figure 3C entirely from the paper, as it is an inappropriate test for the reasons described in the last review (and which the authors acknowledge). Instead, they should put Figure 3-supp Figure 1 in its place.

Original comment 6: The new discussion is a good addition. The authors should add Senzai and Buzsaki (2017) PMID: 28132824 and GoodSmith et al. (2017) PMID: 28132828 in the list of references with Bui et al. and Huang et al., as these papers were the first to show clearly that identified mossy cells had spatial tuning.

[Editors' note: further revisions were suggested prior to acceptance, as described below.]

Thank you for resubmitting your work entitled "Distributed subthreshold representation of sharp wave-ripples by hilar mossy cells" for further consideration by *eLife*. Your revised article has been evaluated by Laura Colgin (Senior Editor) and a Reviewing Editor.

The manuscript has been substantially improved by your efforts but there are some remaining issues that need to be addressed, which can be summarized as follows:

1. Limitations about the identification of mossy cells should be noted. 2. Consider adding experiments with DCG-IV. 3. Revise the significance statement for clarity. Detailed recommendations are below.

*Reviewer #1 (Recommendations for the authors):*

Again, the authors have improved the paper with their revisions.

These residual concerns remain:

1. Identifying mossy cells electrophysiologically and morphologically.

The authors say that cells that "…had a membrane capacitance higher than 45 pF and spines and thorny excrescences on proximal dendrites …are electrophysiological and morphological markers of MCs."

A cell capacitance over 45 pF is insufficient to define MCs electrophysiologically because it is also a characteristic of other cells. Spines do not characterize MCs because they are present on other cell types. Thorny excrescences do but all cells were not filled by biocytin (and we don't know how many were out of all those included).

Therefore, please add a limitation to the text that the approach may have included other cell types, but the authors do not think this is a serious concern because the cells they filled showed thorny excrescences.

2. Regarding the evidence that CA3 SPWRs cause MC depolarizations

The authors say that they have shown that synaptic transmission to the MC via mossy fibers in not affected by SPWRs. However, that was not the point of the suggestion. The point was that blocking mossy fiber transmission is a useful tool to reduce the non-CA3 input. If experiments were done in this condition, it would be more convincing if a SPWR caused a depolarization in MCs because the input that is not from CA3, the majority, would be removed.

3. Regarding the Impact statement that was unclear

It is still unclear. Why is it important that 30% of SWR waveforms were reconstructed. How would a synaptic response of a MC reconstruct a SPR? How would 5 MCs do this?

Is this the meaning? Machine-learning algorithms combined with whole-cell patch-clamp of MCs and LFP recordings of SWRs showed MC EPSPs follow approximately 30% of SWRs?

---

## [Author Response]

Essential RevisionsReviewer #1 (Recommendations for the authors):Detailed additional concerns.A ref is needed for CA3 neurons send input to MCs. I would suggest PMID: 7884451; PMID: 9157312 Note CA3 also sends input to hilar and GCL interneurons. For that I would cite PMID: 8008190, PMID 7473243, PMID: 8300905, PMID: 2358523.

Thank you for your suggestions. We have added some of the suggested references and incorporated them into the text in the Introduction and Discussion sections.

“MCs receive synaptic inputs from CA3 pyramidal cells (Scharfman, 1996, 1994), and their axons send synaptic outputs to DG granule cells (Buckmaster et al., 1992).”(Page 3, Line 61)

“note that it is known that there are projections from the CA3 region to inhibitory interneurons in the hilus (Kneisler and Dingledine, 1995).” (Page 13, Line 387)

Line 62"there is no direct axonal projection from CA3 pyramidal cells to DG granule cells Fujise and Kosaka, 1999"This is not likely to be true. Li et al. (PMID above) showed CA3 axons in the GCL and IML. However, I do not believe synapses were proven. Still, it is highly suggestive. Buckmaster showed similar findings in primate. PMID: 11135261

Thanks for the insightful comment. We rewrote the sentence as follows:

”While it has been shown anatomically that axons from CA3 pyramidal cells project to the inner molecular layer and granule cell layer, it is not clear whether they actually form synapses and serve their function (Li et al., 1994; Vivar et al., 2012).” (Page 3, Line 63)

Line 63"An estimated 15,000 MCs reside in the rat hippocampal formation, whereas approximately 300,000 and 2,450,000 CA3 pyramidal and granule cells exist, respectively"I am not sure these numbers are correct. As a start, see PMID: 17765743, PMID: 17475251 Perhaps the authors are referring to a particular species and if so that should be discussed. Finally, estimating GC numbers is very difficult because they overlap even in thin sections. So this needs to be considered carefully. See PMID: 29311853, PMID: 1793176, PMID: 3292009

Thank you for providing us with the references. We have carefully examined the references and other relevant papers, reviewed the MC and DG cell numbers, and revised them as follows:

“An estimated 30,000 MCs reside in the rat hippocampal formation, whereas approximately 300,000 and 1,000,000 CA3 pyramidal and granule cells exist, respectively. ”(Page 3, Line 67)

Since it is not clear what the numbers are, it might be useful to show a range, and implement the deep learning with several possible values. If the results are the same that would make the results more convincing.

The neural network in this study, predictions are made using recorded MC membrane potentials, so the total number of neurons in each area does not affect the results or interpretation.

Line 66. Same caution needs to be made here

After making changes to the number of cells, the following sentences were determined not to require any changes:

“Thus, the number of MCs is one to two orders of magnitude lower than that of CA3 pyramidal neurons and DG granule cells (Figure 1 —figure supplement 1), and MCs constitute a small relay layer.” (Page 3, Line 69)

Line 103I think the authors mean cells were excluded if capacitance was <45pF (please explain the reason) and they were included if they had spines and thorns.

MCs are typically characterized by having a large soma compared to surrounding cells (Amaral, 1978). Since membrane capacitance is proportional to soma area, we adopted a criterion based on previous studies (Hedrick et al., 2017). In addition, the wording in the sentence has been rewritten as follows:

“Data were included in the analysis if the recorded cells had a membrane capacitance higher than 45 pF and had spines and thorny excrescences on their proximal dendrites.” (Page 20, Line 631)

Refs are needed that these are characteristics of MCs.

Thank you for the suggestion. The papers Amaral 1978 and Frotscher et al. 1991, which report that MC has thorny excrescences, were added to the references.

Line 109Consider excluding cells with such instability that they were not held >1 min.How were others designated healthy?

Although we describe in the paper the actual recording time for which data were obtained as 55 s to 357 s, this does not mean that the recordings are unstable, as we sometimes interrupt the recording to check for changes in access resistance. As described in Methods, we included data when the mean resting membrane potential of each MC was < -50 mV and Z-scores of the mean membrane potentials for 30 s were between -2 and 2. The latter was done to exclude cells with large membrane potential fluctuations, i.e., unhealthy cells.

Line 465Exposing mice to an enriched environment means what?Why was this done? What were the effects? Was there a control?

We apologize for not providing a reason to expose mice to an enriched environment. The reason is to increase the occurrence of SWR during experiment. Our recent paper showed that short-term preexposure to enriched environment augments hippocampal ripple (Okada et al., 2024; PMID 38797694).

The following sentence was added to the method section:

“Before the surgery, the mice were exposed to an enriched environment for 30 min to increase the occurrence of SWR during the experiment.” (Page 20, Line 639)

Line 479Why was -40 a cut off? That seems very depolarized and unhealthy.How were MCs distinguished from CA3 or GCs? What about slow spiking hilar interneurons?What were the brief inward currents exactly?

Criteria were not only membrane potential, but also access resistance and stability of resting membrane potential. This recording was a blind patch located deep in the hippocampus, making targeting difficult, and was used if the membrane potential was stable during the recording.

All in vivo patch-clamp recording data were reconstructed by injecting biocytin during the recording to confirm that the recorded cells existed in the hilus. In addition, electrophysiological properties differentiated the cells from granule cells and other interneurons by confirming the presence of sags and that they were regular spiking cells. (Scharfman 2016, Nat Rev Neurosci)

To identify the firing properties of the neuron, 500-ms current from -100 to +100 pA were injected into the neurons at steps of 20 pA.

Line 480What was the tungsten electrode exactly? What part of CA1 was recorded? Why? To record SW? If SW were recorded in CA1, the layer should be consistent. Was it checked?Exactly how was DiI used?Why were SW recorded in CA3 in vitro and CA1 in vivo?Was the site in CA3 in vitro consistent across slices? How was it positioned so it was consistent?Why is the term SW used, not SWR?

In an in vivo experiment, SWR was recorded from the striatum oriens to the radiatum of CA1 using a tungsten electrode.

DiI was dissolved in acetone and methanol (1:1) at a concentration of 40 mg/ml. The electrode was directly immersed in the dissolved solution for several seconds to allow the DiI to adhere.

Because of the anesthetized and blinded recording, it was difficult to combine the patch-clamp recording from MC with the LFP recording from CA3 in vivo. Therefore, we note the difference in interpretation of results obtained in vitro and in vivo (Page 12, Line 357).

Yes, the recording site in CA3 in vitro was consistent across slices. The CA3 recording site was always located in the CA3 pyramidal cell layer closer to the hilus, which is the inner part of the CA3 pyramidal cell layer, assuming that a straight line connecting the suprapyramidal edge and the infrapyramidal edge.

Sorry for the confusion with the terminology, SW has been reworded to be consistent with SWR.

Line 503'The detected events were scrutinized by eye and manually rejected if they were erroneously detected "Please explain.

Basically, we automatically detected SWR peak times that were determined at a threshold above the mean + 5×SD of the baseline noise by using Matlab. However, to confirm that the detection system was working properly, we finally checked the exact SWR events that were detected. Even with the criteria, noise could still be detected as an event, in which case it was manually rejected. Previous study also used a similar process, thus it is not an eccentric method (Mizunuma et al., 2014).

Line 500Were definitions of SW similar in vivo and in vitro? Were the conclusions from SW in vitro confirmed by SWs in vivo? The data seem mainly from slices.

We do not think that the definition of SWR in vivo and in vitro is strictly the same, because in vivo experiments are under urethane anesthesia. However, as analyzed in Figure 3E, the in vitro data are supported by the fact that even the in vivo single MC data significantly predict SWR from membrane potentials of MC when compared to surrogate data. Even with the limitation of being under anesthesia, the fact that SWR could be predicted is useful because it shows that the prediction model can be applied to in vivo data.

Please explain deep learning in more detail. One has to read the results to infer how this was done and many other aspects of the methods. The figure legends are not very helpful in this regard, making it hard to read the paper.

Thank you for your suggestion. In response to reviewer #2’s detailed comments, details of the neural network have been added to the method and legend.

Please explain in the methods what the authors mean by:spatial entropyRMSEMDS spacebinarized predictionsignificantly predictiveSW identity numberchance level of spatial entropySW permeation by MCs

We clarified the meaning of each word and added it to the result, legend, or method.

Line 562How was the subgranular zone defined?Consider a supplemental figure showing the method.

After Nissl staining for recorded slices, the relative location of MCs is calculated in ImageJ. The subgranular zone was defined as the innermost layer of the granule cell layer where cells are densely distributed. Detailed information on this method is provided in the previous publication (Koyama et al., 2012).

If slices were transverse, the DG is not a semicircle. If horizontal sections were made, it is not either. Therefore explain how the DG was simplified to a semicircle.How was the edge of CA3 defined without a stain?

After recording, the edges of the DG and CA3 can be identified by staining Nissl, as shown in the bottom of Figure 1A. The brains were horizontally sliced at an angle of 12.7°, and although the DG is not exactly a semicircle, the positions of the both edges of the DG and the edge of CA3 were used to normalize the figure as a semicircle.

StatisticsPlease add means and SEMs where n is the number of mice. This could be done in supplemental material.

Thank you for your suggestion. Detailed information for each slice, such as the number of mice, the number of SWRs, and the recording time, are listed in Supplementary File 1.

Stats are not explained at all. This is a serious omission. For example, what were the parametric and non parametric tests, and what were the tests of normality to make these decisions? Or was it assumed all data were normal? Were ANOVA's two-way and if so were there interactions?

We have performed a non parametric test because our data is not normally distributed, and we have not performed a two-way ANOVA.

We have added the following sentence to the methods:

“Since the data did not follow a normal distribution, a non-parametric test was performed.” (Page 25, Line 785)

Figure 1:SW are not clearly SW. The traces are too compressed to tell. What are the criteria in vitro?What are the criteria in vivo? If animals do not have immobility but are forced to be immobile, what relationship are the SWs of slices and anesthetized animals to SW in vivo in awake behaving animals?For the expanded traces in Figure 1B what is the voltage calibration?

Sorry for the lack of details in the method. We have added the in vitro and in vivo criteria in the "SWR detection" section of the method.

Thank you for your insightful question. We have added the following sentence to the discussion:

“In this study, we performed MC patch-clamp recording both in vivo and in vitro, and clarified that SWR can be predicted from *V*m of MC in both cases. However, there are three caveats to the interpretation of these data. First, the in vivo SWR cannot be said to be exactly the same as the in vitro SWR: note that in vitro SWR has some similarities to in vivo SWR, such as spatial and spectral profiles and neural activity patterns (Maier et al., 2009; Hájos et al., 2013; Pangalos et al., 2013). The same concern applies to MC synaptic inputs. The in vivo *V*m data may contain more information compared to the in vitro single MC data, because the entire projections that target MCs are intact, resulting in a complete set of synaptic inputs related to SWR activity, as opposed to slices where connections are severed. While we recognize these differences, it is also very likely that there are common ways of expressing information. Second, since the in vivo LFP recordings were obtained from the CA1 region, it is possible that the CA1-SWR receives input from the CA2 region (Oliva et al., 2016) and the entorhinal cortex (Yamamoto and Tonegawa, 2017). In addition, urethane anesthesia has been observed to reduce subthreshold activity, spike synchronization, and SWR (Yagishita et al., 2020), making it difficult to achieve complete agreement with in vitro SWR recorded from the CA3 region. Finally, although we were able to record MC *V*m during in vivo SWR in this study, the in vivo data set consisted of recordings from a single MC, in contrast to the in vitro dataset. To perform the same analysis as in the in vitro experiment, it would be desirable to record LFPs from the CA3 region and collect data from multiple MCs simultaneously, but this is technically very difficult. In this study, it was difficult to directly clarify the consistency between CA3 network activity and in vivo MC synaptic input, but the fact that the SWR waveform can be predicted from in vivo MC *V*m in CA1-SWR may be the result of some CA3 network activity being reflected in CA1-SWR. It is undeniable that more accurate predictions would have been possible if it had been possible to record LFP from the CA3 regions in vivo.” (Page 12, Line 357)

The inset of 1B has been magnified. Since only the time width is enlarged, the scale bar is omitted.

The traces need to be decompressed to tell when during the SW the MC activity starts. That is important so that one can tell if it was possibly monosynaptic or polysynaptic. However, it is recognized that with field potentials the onset of CA3 activity will be hard to know precisely. Therefore monosynaptic CA3-MC connections will be hard to confirm. If polysynaptic circuits are involved this is very important because it might be CA3 only activates MCs when secondary pathways become involved. Importantly, polysynaptic transmission is always variable compared to monosynaptic transmission. Together there are many reasons MC responses to SWs are variable as discussed above.

Thank you for raising a very good point. We have added the following sentence to the discussion:

“The synaptic transmission to the MC is also influenced by many factors not only from the CA3 region, such as input from other hilar interneurons, semilunar granule cells, and GC (Larimer and Strowbridge, 2008; Larimer and Strowbridge, 2010; Scharfman and Schwartzkroin, 1990); note that it is known that there are projections from CA3 region to inhibitory interneurons in the hilus (Kneisler and Dingledine, 1995). Conversely, in the case of MC, where there is much spontaneous activity, the fact that the SWR can be predicted from the shape of the EPSP at the time of SWR occurrence suggests that the information of the SWR is strongly reflected in the synaptic inputs of MC.” (Page 13, Line 383)

If MCs have different resting potentials in Figure 1B it would seem logical and not very informative to point out that their depolarizations in response to a SW are variable. An analysis of variability as it relates to resting potential could help. One reason is at depolarized potentials NMDA receptors will become important and broaden EPSPs of MCs.If it is variable in the same MC at the same resting potential, that is nice but it is (1) not clear how much this has been observed (2) if that is surprising since SWs themselves are variable.

Thank you for the comment. Even if the depolarization in response to a SWR varies between cells, the depolarization reflects the input from upstream neurons, therefore we think it is important that even a single cell has enough information to predict the SWR. Please also note that diversity and heterogeneity in physiological characteristics, as well as at the molecular and genetic levels, are known to be important in information representation (PMID: 20802489; 23630284).

As noted above, these have been added to the discussion.

Figure 1 and elsewhere. Please add number of mice throughout.

This point was described in the original manuscript. (Page 5, Line 115)

I can't see traces in C. Here and elsewhere they are too small. Please explain MDS1 and MDS2. Furthermore, what are the traces from? What does the sphere of dots signify?

The waveform in Figure 1C is magnified. The original waveforms are inserted with data reduction so that the SWRs that are not similar in the MDS space are placed far away from the original waveforms. The multidimensional scaling (MDS) method is an analytical technique that transforms the relationship between objects into a visually comprehensible form (typically reduced data to two or three dimensions) based on similarity. The RMSE (i.e. dissimilarity) between each SWR waveform is calculated, and based on this value, each SWR waveform is plotted in a two-dimensional array by the MDS algorithm.

D. Based on the comments above, consider a plot of depolarizations that Is just for cells at a small range of membrane potentials to see if there is more consistency. Also consider other factors: latency from the onset of the SW, duration, area under the curve. Are these just as a variable or not? If in variable it would be insightful. Can variability be reduced by use of APV to block NMDA receptors, cutting away CA3a so only CA3b and c are present, or just c, etc? One could also block GC transmission with DCG-IV and GABAergic transmission to reduce polysynaptic influences.

As shown in new Figure 1 —figure supplement 2, the EPSP peak occurs 0.038 s after the onset of the SWR, therefore there is almost no fluctuation. As for duration and AUC, the EPSP is often seen in a barrage-like pattern, so it is considered inappropriate for further analysis.

In D, what is Vm# on the Y axis? Was this just done for the one slice with 5 MCs? What about other data? Was it reproduced? What was the finding here? Just that there is variability?

The y-axis shows the membrane potential response of the MCs to each SWR. As you indicated, Figure 1D was created using one slice with quintuple recording as a representative. We have confirmed that there is heterogeneity in the membrane potential response for the other slices as well. Yes, as you say, this figure shows that there is variability in the MC response to SWR.

E. Why z standardize? What were the weak correlations (I just see one R value only) and were they statistically significant? what was the test?

Since the baseline SWR amplitude or ΔVm varies from slice to slice, the Z-score of each value between slices was performed to allow relative evaluation. The R values suggest that there is at least a weak correlation of ΔVm with SWR amplitude.

Why are data from MCs pooled? In other words, why equate the 87 cells if they were from 23 slices and even fewer mice?

Thanks for pointing out a good point. On average we can obtain 7 to 8 slices per mouse. As you pointed out, of course there are differences between different cells, slices, and animals. However, we kept the experimental conditions constant, such as the age of the mice and the recording site of the LFP. Although we were able to confirm differences in the accuracy of the predictions for each cell, slice, and animal, since the accuracy of the predictions for the real data was better than that for the shuffled data in 75.9 % of the cases (Figure 3 —figure supplement 7), we can conclude that the results obtained from this model generally have common properties. On this basis, we believe that there is no problem in pooling the z-scores of the recorded SWR and the membrane potential of the MCs in response to the SWR and deriving an overall trend in the response of the MCs to the SWR.

Here and elsewhere there are numerous impressive recordings from more than 1 MC. However, how are the relationships interpreted in MCs of the same preparation if sometimes there are 1, 2, and other times up to 5? One wonders because totals are often cited as if everything is pooled. Moreover, there is a correlation between the number of cells recorded in the same preparation and some outcomes. Because of that finding, I don't see that pooling is justified.

Since our study uses neural networks, it would usually require a large amount of data. We standardize each data set by performing Z-scoring within each data set or each MC, and then perform pooling. Moreover, we have checked the differences in prediction accuracy for each MC, slice and animal as mentioned above (Figure 3 —figure supplement 7). For more information on the statistical significance of Z-scores, please refer to the following references. (DOI; https://doi.org/10.1007/978-3-031-33837-3_8)

Figure 2. In B there is no GluR2/3 stain in the inner molecular layer where MC axons are located.

GluR2/3 is used as a MC marker, and is generally identified as MC by checking its expression in the soma of MCs, without checking staining in the inner molecular layer. See Danielson et al., 2017 (PMID: 28132825)

Why was 100-250 Hz used as a filter and what was the type of filter? Presumably this is to study ripples. But ripples are not typically considered over 200 Hz and below 100 Hz is often included. Also the figure shows ripples besides the two SW events- what were these- small SWs?The inset needs a voltage calibration.

We used a band pass filter to extract specific signals, in this case 100-250 Hz. Previous studies have shown that the presence of high power in the ripple band itself is often used as a marker for SWRs, and the different frequency bands are used for each study. However, the characteristics of spiking and LFP activity observed during SWRs are consistent across a variety of studies using different thresholds (Joo and Frank, Nat Rev Neurosci 2018). Therefore, we are of the opinion that the 100-250 Hz is not inappropriate for the consideration of ripples.

Although the small SWR next to the two SWR events was also detected as SWRs, we did not select it for the representative event.

The voltage calibration in the inset is the same as left traces.

Figure 3. In A, the neural network is not explained thoroughly. What were the assumptions based on? What does a full connected link mean relative to one that is not connected? Why is there only unidirectional flow in light of the complex circuitry? What is the take home message in B, F? In C, what were the statistics? Please make symbols in D large enough to see. Explain the box and whisker plots. Were these data from one slice? Were they reproduced?Please explain E.

As we wrote in the main text, our neural network was inspired by the autoencoder structure. We constructed a simple neural network which enables us to reconstruct the SW waveform in CA3 from the membrane potentials of MCs. Because the purpose of the model is to explore how backprojecting CA3 SWRs are encoded by the MC activity, we constructed a unidirectional model that predicts the SWR waveform from the membrane potential of the MC. We did not propose that this autoencoder is implemented in the actual brain; rather, we used it as a tool to assess the extent of information about SWRs contained within the MC populations.

A ‘full connected link (layer)’ is a layer in a neural network in which every neuron in the layer is connected to every neuron in the previous layer. This layer is the most fundamental layer of the multilayer perceptron and often used in Convolutional Neural Networks.

In B and F, we show that in both in vitro*/*in vivo, the waveform output from the real data set is closer to the original than the waveform output from the shuffled data set. In addition, in C, KS test shows that the RMSE is calculated as the prediction error from the original waveform, and that this error is reduced when the real data set is used.

For D, we modified the plot by darkening the color to make it easier to see.

The box plots were obtained from all of the slices used in the paper. The data was Z-scored within datasets. For example, from the data with 5 cells recordings, we also obtain the results (RMSEs between original and predicted by real input) with a combination of 1, 2, 3, and 4 cells and these values are grouped together and Z-scored, and plotted as box plot. Because the feature of SWR differs from dataset to dataset (for example, in dataset1, SWR is generally low amplitude, in dataset2, SWR with high amplitude is often seen, etc.), we think it is appropriate to Z-score within the dataset.

Figure 4 Please explain the title. In A, what is the count of? In C, please explain the method of analysis of spatial entropy. The methods are not clear and the relationship of what is said in the methods about entropy to Figure 4C is not clear.

Since it was found in Figure 3 that it is possible to predict some SWRs from a single MC, we focused on SWR prediction from a single MC in Figure 4, showing a histogram of how well each MC can predict SWRs (Figure 4A) and examining whether the distribution of MC locations and electrophysiological characteristics is related to its predictability (Figure 4B-D).

Sorry for the lack of clarity for the methods about calculating spatial entropy in Figure 4C. We have inserted the following details to the Methods section for “Spatial entropy”:

“Spatial entropy was calculated to investigate whether the distribution of recorded MCs in the semicircular diagram is related to the predictability of SWR (Figure 4B). The spatial entropy for the semicircular diagram was calculated by converting the polar coordinate values to Cartesian coordinate values and calculating the Euclidean distance between each two points. This resulted in an average of five % predictable SWRs for each MC and its four nearest neighbors in the semicircular space. This procedure was repeated to collect averages for all MCs. The spatial bias of the average % predictable SWR was then quantified by spatial entropy. Spatial entropy was calculated as -ΣP*i* log2 P*i*, where π is the probability frequency of the mean % predictable SWR of the MC with MC identification number *i*. The chance level of spatial entropy was estimated by 10,000 surrogates in which the value of % predictable SWR was permuted across the MCs. When the spatial entropy is significantly lower than the shuffle, the MCs that are more likely to predict SWR indicate a bias in anatomical location. ” (Page 24, Line 751)

It seems here that it does not matter where the MC is located. Is that the message?

This point was described in the original manuscript. (Page 8, Line 221)

In D, E how was the single SW selected?

Sorry for the mistake in the legend. The correct text is as follows:

“Each dot indicates a single MC.”

In Figure 5, What is "predicted deflection"? "dimensionally reduced"? What is the threshold for defining a SW event was predictable in A?

Sorry for the confusing text. For “predicted deflection”, we have rephrased to “biased prediction” which means that there is a bias of predictable SW within each MC (Page 8, Line 230). For “dimensionally reduced”, we plotted the recorded SWR reduced to two dimensions by the MDS method. As we wrote in the main text, if the RMSE predicted by the real data sets was significantly lower than that predicted by the shuffled data sets, the SWR was defined as ‘predictable’ by the MC.

B. What is the spatial bias of a rate? Change distribution? Are "surrogates' the predicted events and non-surrogates are the actual SWs? What is the bottom line of this figure?

We show in Figure 1C that the plots of SWRs reduced to two dimensions by the MDS method are SWRs with more similar characteristics the closer the plots are. In other words, if the distance between two plots is close, the features of these two SWRs are similar, and if the distance is far, the features of these two SWRs are completely different. Therefore, we asked whether the predictable SWR depends on the SWR features (i.e., whether they are clustered), and the spatial entropy was calculated to check whether the predictable SWRs (red) are partially agglomerated. If it does not depend on the SWR features, then the entropy will be large because the predictable SWRs (red) are distributed without bias. If the question is correct, the entropy will be smaller in the actual data than the shuffled positions of the red predictable SWRs (change distribution). In our data, 37 out of 87 cells of all MCs showed that the predictable SWR depends on the SWR characteristics, i.e., a particular MC is associated with a particular SWR cluster.

The very intriguing hypothesis in the abstract – i.e., a particular MC showed a more robust association with a particular SWR cluster, and the SWR cluster associated with one MC rarely overlapped with the SWR clusters associated with other MCs. – where is the evidence supporting this hypothesis? It seems only based on the complex analysis where there is a model driving the conclusions and the model may not be a good model.

A particular MC showed a more robust association with a particular SWR cluster is explained in Figure 5 as commented above. The SWR cluster associated with one MC rarely overlapped with the SWR clusters associated with other MCs is the interpretation from Figure 6.

Line 152. The authors find that the prediction error of real data was lower than shuffled data. Then they conclude from this that CA3 "information during SWs was at least partially preserved in the Vm responses of MCs" I see this as a major leap. All I see is that the deep learning had less of a prediction error for real data than shuffled data. Why is there an inference that this is meaningful in the way the authors state?

We referred to the fact that information in a given SWR is reflected in and coupled to the membrane potential of the MC as “information during SWRs was at least partially preserved in the Vm response of the MCs”. If the information in the SWR is not reflected in the membrane potential of the MC and the two are unrelated activities, then the SW waveform should be largely unpredictable from the membrane potential of the MC. In fact, as shown in Figure 3B,C, the SW waveform output with shuffled data as input and the SWR waveform output with real data as input have significantly different prediction errors (RMSEs) from the original SWR waveform.

Line 160. The MCs were divided into subsets. Please explain what was done exactly. Why is it surprising that the more MCs are recorded simultaneously the better the prediction can be?

For example, in the quintuple patch-clamp recording, all 5 cells are used for training. In addition, 4 cells (or 1 cell) can be extracted from the 5 cells with 5 combinations can be trained. Similarly, 3 cells (or 2 cells) can be extracted from 5 cells with 10 combinations can be trained. A similar procedure is performed with the quadruple, triple patch-clamp recording dataset.

Interpreting the results in Figure 6, the encoding of SWR by MCs is not perfectly orthogonal, and the predictable SWR may overlap between MCs recorded at the same time. In other words, it is expected that the prediction accuracy of SWR waveforms in neural networks can be improved when several different MCs encode the same SWR. Therefore, the result that prediction accuracy increases when many MCs can record simultaneously is consistent with the results of single-cell MC analysis.

Line 169. Here the frequency range is 120-250 but in the figure it is 100-250.

Sorry for the confusion. We have corrected to “120-250 Hz” in the Figure 3 —figure supplement 6

Line 172. Please explain why the data suggest MCs "retain more information about SWRs"

The more MCs, the closer to the original waveform, including not only the RMSEs but also the high-frequency details, which means that the more MCs, the more accurate the reproduction of the SWR. We have expressed this as “retain more information about SWRs”.

Line 182. The SW waveform was predicted by the MCs but I am not sure that is interesting because the traces show that SWs are so similar from one example to the next. So it would be easy to predict.

Indeed, we agree that each SWR looks similar. However, in our experience, predicting SWR is not easy. Because there are many factors that make SWR different, such as amplitude, duration, ripple power, etc. As a result, it was not possible to predict the SWR waveform close to the original using conventional linear regression, taking into account these factors. Thus, instead of predicting similar average waveforms, we constructed a new neural network system that could significantly reproduce the characteristics of each individual SWR based on the Vm information of the MC. We believe that the novelty lies in the fact that we were able to link upstream neural dynamics and downstream neural response using our originally constructed neural network system.

Line 205. Only two intrinsic characteristics of MCs were examined so it seems hard to conclude that characteristics of MCs do not matter.

We agree with the reviewer's comments. Therefore, we performed calculations of the average amplitude and frequency of EPSPs during the recording time as a measure of spontaneous activity of MCs. We found no correlation between the percentage of predictable SWR and EPSP amplitude, but a negative correlation between EPSP frequency and SWR (Figure 4 —figure supplement 9). We have added following sentences in the Results section:

“We also examined the correlation between percentage of predictable SWR and EPSP of MCs (Figure 4 —figure supplement 9). We found no correlation between EPSP amplitude and SWR, but a negative correlation between EPSP frequency and SWR, suggesting that the higher the activity of the MC, the more difficult it is to predict SWR, and that less active MCs predict more accurate SWR. ” (Page 8, Line 226)

The fact that MCs in the area of the lower blade were more predictive could simply be due to greater connectivity with CA3 in that location. It would be good to investigate.

This result was unexpected for us, as we responded to Reviewer #2's point that we are not consistent in our in vivo data due to the low cell counts. As we cannot rule out the possibility that this may have been influenced by the preparation of acute slices, we will need to carefully examine this in future research.

Line 211 What is meant by a dimensionality of SWs and reducing a SW to two dimensions?

Here, the SWs predictable from each MC are reduced to two dimensions to verify whether they are clustered, making them visually easier to understand. The following sentence has been added to the Method:

“To ensure that similar SWRs are placed close together and to visualize them in two dimensions, the MDS algorithm was used in this study. The plot reduced to two dimensions is referred to as MDS space. The variable used in the MDS algorithm is the matrix of RMSEs (i.e. dissimilarities) between simultaneously recorded SWRs.” (Page 23, Line 731)

The paragraph starting on page 224 is critical to the conclusion that is novel in the paper but it is not well explained. Therefore it is hard to be convinced. For example, is it justified to binarize prediction scores into 1 and 0? Is it reasonable to pool data from all MCs whether they were recorded with 1 other or 2 or more?

We apologize for the lack of explanation. Setting the predicted score to 1 or 0 reduces the resolution, but does not change the results of the analysis. We also calculated 113 pairs of two MCs, all for simultaneously recorded MCs and SWRs. Since we did not perform calculations for MCs that were not recorded at the same time, we consider it reasonable to pool them.

Reviewer #2 (Recommendations for the authors):(1) Concerning more detailed model descriptiona – Details on model architecture: The manuscript briefly mentions the use of an encoder-decoder architecture with dense layers, but the rationale for choosing this architecture or the specific configuration of the layers used is not fully explained. The inclusion of such details, along with the rationale for the choice of activation functions (relu and sigmoid), would provide deeper insights into the model design.

We thank the reviewer’s comment. The rationale for selecting this model architecture is added in Discussion:

“Specifically, although waveform prediction was possible using simple linear regression methods such as linear regression and ridge regression based on RMSE evaluation, it was not possible to reproduce the high frequency band, which is an important feature of SWR. … Using this simple neural network, we were able to predict LFP waveforms more accurately than chance prediction from the *V*m dynamics of 1-5 MCs, and were also able to reproduce the high-frequency band of the SWR.” (Page 11, Line 322)

The activate function has been set with reference to autoencoder code from The Keras Blog (https://blog.keras.io/building-autoencoders-in-keras.html) and added to Methods:

“The activation function was set to *relu* when compressing from encoder, and the activation function was set to *sigmoid* when restoring from latent variables.” (Page 22, Line 700)

b – Details on the training process: The paper would benefit from a more thorough explanation of the preprocessing steps applied to the data prior to training, as well as the rationale for the chosen loss function (root mean squared error, RMSE) and the detailed settings of the optimizer (e.g. learning rate, β values). This information is crucial for anyone attempting to replicate the study or apply the methodology to similar problems.

Thank you for your comment. We have added following details on the training process to Methods:

“Before training, the traces of the *V*ms and SWR waveforms were preprocessed by scaling them between 0 and 1. ” (Page 22, Line 691)

“The network was optimized by adaptive moment estimation (Adam) with a learning rate of 0.001. The parameters for the optimizer Adam were as follows: β1 (an exponential decay rate for the first moment estimates) = 0.9, β2 (an exponential decay rate for the second moment estimates) = 0.999 and ε = 10^-7^; the default values were used for the other parameters. ” (Page 22, Line 702)

c – Details on data handling and evaluation metrics: Although cross-validation is mentioned, a clearer explanation of how the data sets were split and the criteria used for this split would be helpful.

Thank you for your comment. We have added details on cross-validation to Methods:

“Each dataset was equally divided into 5 subsets in order of occurrence so that all SWRs would be test data at least once; one subset was used as the test data, while the remaining 4 subsets were further divided into 10 subsets (one subset was used as the validation data, and the remaining 9 subsets were used as the training data). If the number of SWRs is not divisible by 5, to keep the number of SWRs used in the test data constant, some of the same SWRs are used as test data in the fourth and fifth rounds of cross-validation.”(Page 23, Line 715)

d – Details on implementation and computational resources: Specifying the versions of TensorFlow, Keras and other dependencies used would help in replicating the computational environment. In addition, information about the required computational resources, such as GPU specifications (if applied), would provide practical insights into the feasibility of reproducing the study.

The following test was added to the “Prediction of SW waveforms from Vms” of Methods:

“A neural network model was designed to predict the SWR waveforms from one to five simultaneously recorded *V*ms using Python 3.8, TensorFlow 2.3.1 and Keras 2.4.3. The GPU specification applied was NVIDIA Quadro RTX 4000.” (Page 22, Line 685)

(2) Manuscript organizationThe figures are effective in presenting the data, but the narrative in the text can be difficult to follow in places. Reorganizing the manuscript could facilitate better comprehension for readers. Specifically:- Methodological descriptions should be confined to the Methods section.- Theoretical and analytical concepts could benefit from succinct explanations where they first appear.- Transitions within the narrative should be smooth and well-motivated. For instance, the spatial analysis on page 7 requires a abrief introduction to provide context.- A noteworthy result regarding the predictive value of MCs in relation to SWR information, currently discussed on page 11 (lines 300ff), could be more appropriately placed in the Results section.

Thank you very much for your detailed instructions on manuscript organization. We have changed the content as much as possible according to your suggestions.

(3) References:a – Hyde and Strowbridge (Nat Neurosci, 2012) have previously published research relevant to the topic; they used artificial stimuli to demonstrate encoding of temporal sequences into the MC network, suggesting mechanisms for short-term memory. The authors should acknowledge this study and discuss these earlier results in the context of their own findings.b – Nitzan et al. (2020) undoubtedly serves as a valid reference for supporting the propagation of ripples from CA3/CA1 to cortical targets. However, since this phenomenon has previously been demonstrated and discussed, it would be preferable to cite earlier studies that originally identified or thoroughly investigated this propagation.

a – Thank you for the insightful comment. We have added to the introduction related to the comment as follows:

“Moreover, it has been shown both experimentally and theoretically that the activities of MCs are involved in memory processing, such as pattern separation (Hyde and Strowbridge, 2012; Myers and Scharfman, 2011). ” (Page 4, Line 85)

b – Thank you for the suggestion. We have cited further references from Siapas and Wilson 1998 and Sirota et al., 2003. (Page 3, Line 79)

(4) Further commentsa – To maintain consistency in the literature, it is recommended that the authors align with established nomenclature. Specifically, 'SW' typically denotes slow waves, while 'SPW,' introduced by Buzsáki (1986) for sharp waves in the hippocampus, and SPW-R (or SWR), for sharp wave-ripple complexes, are the accepted terms. Using these conventions would enhance clarity and ensure alignment with the field's standard terminology.

Thank you for your insightful comment. SW has been reworded to be consistent with SWR.

b – "Input" versus "Response": The manuscript's language at times implies an active role of mossy cells. However, the analysis primarily concerns *synaptic inputs* from CA3 to MCs, indicative of upstream network dynamics, rather than direct responses (that is, spiking activity) of MCs. Since responses of MCs in the true sense were not the focus of this study, I recommend a thorough review and revision of the manuscript to accurately reflect this distinction.

Thank you for your suggestion. We fully agree with this comment. We have corrected the ambiguous description by clearly stating that the membrane potential response of the MC is a synaptic input from the CA3 region.

c -Line 222f: "… a MC tended to predict a specific subset of SWs …" – A more appropriate (precise) formulation would imply that it is not MCs that predict a specific subset of SPWs, but the synaptic input that an MC receives from the upstream network.

We thank the reviewer’s comment. We have rephrased the sentence to “… a synaptic input to the MC received from the upstream network tended to predict a specific subset of SWRs with similar waveforms.” (Page 9, Line 246)

d – Line 293: "Only a single MC was recorded in our in vivo study." – This sentence seems to be misleading: If I am not mistaken, the authors meant to write that their in vivo dataset consists of single MC recordings, as opposed to their in vitro dataset, which contains recordings of up to five MCs recorded simultaneously. Please rephrase this sentence.

We thank the reviewer's comment. We have rephrased the sentence to “Finally, although we were able to record MC *V*m during in vivo SWR in this study, the in vivo data set consisted of recordings from a single MC, in contrast to the in vitro dataset.”(Page 13, Line 372)

e -Line 296: "…because the nerve fibers are preserved without sectioning in the in vivo preparations." – This statement, while not incorrect, is somewhat unspecific; the authors should perhaps specify that in vivo, CA3 projections targeting MCs are intact, resulting in a complete set of synaptic inputs related to CA3 network activity, as opposed to slices where connections are severed.

We thank the reviewer's comment. The following has been rewritten based on the reviewer’s opinion.

“The in vivo *V*m data may contain more information compared to the in vitro single MC data, because the entire projections that target MCs are intact, resulting in a complete set of synaptic inputs related to SWR activity, as opposed to slices where connections are severed;” (Page 12, Line 363)

f -Line 313: "…the exponent α was 0.920 with a 99% confidence interval of [0.907,0.933]." – Please check the mathematical terminology used in the description of your equation. In the context of the given equation p=1−α^n, α serves as the base of the exponential term, and n is the exponent. Please correct.

We appreciate the reviewer's comment. We agree with your point and have rephrased to "…the base of the exponential term α was 0.920 with a 99% confidence interval of [0.907,0.933]." (Page 10, Line 281)

g -Line 633: "… (in vivo) n = 273 SWs from 2 quintets." – Single MCs were recorded in vivo. Please correct.

We apologize for the confusing text. Since the text seems to have been moved to an inappropriate place, we have corrected it.

h -Figure 2: The inset in panel B does not convincingly show the thorny excrescences. Can the authors add a microscopic image that shows this crucial feature of MCs more clearly?

We apologize for presenting images that are difficult to understand. Regarding the enlarged image, a higher resolution version has been posted. (Figure 2B)

i -Figure 3: Scale bars should be added to the traces displayed in panel B and F.

We thank the reviewer’s comment. We have added the scale bar for the time scale (horizontal axis). Since the vertical axis of each waveform is scaled to a value between 0 and 1 as a preprocessing step, all output waveforms are also scaled. Therefore, the vertical axis scale bar is omitted. (Figure 3B,F)

j -Figure S3: "Reproducibility of high frequency (120-250 Hz) increases…" – A more precise title would be "Reproducibility of 120-250 Hz coherence increases…"

We thank the reviewer’s comment. We have rephrased to “Reproducibility of 120-250 Hz coherence increases as the number of MCs increases”.

k -Figure S4: The source of the predicted sharp wave/ripples (SWRs) is indicated as being from the CA3 region, yet the validation of prediction errors appears to be based on data from the CA1 region (main Figure 3). Please review and correct, if applicable.

We apologize for the mistake in the figure. We have corrected the word “CA3” to “CA1”.

Reviewer #3 (Recommendations for the authors):1) Lines 69, 324: SWs do not propagate from CA3 to DG (other than volume conduction). Rather, the SW is a reflection in the LFP of local activity at the cellular level (spikes and synaptic inputs). It is the action potentials from CA3 cells that propagate the signal to drive mossy cells. I know what the authors mean, but it is important to be precise in one's language.

We totally agree with your comment. We have rephrased each as follows:

“Consistent with these anatomical features, sharp wave-ripples (SWRs) may propagate from the CA1 region to the DG via synaptic activity in the MCs.” (Page 3, Line 72)

“SWRs originating in the CA3 region are likely to be reflected in the MC as synaptic activity and transmitted to the DG.” (Page 14, Line 409)

2) Line 72: SWs occur "primarily" during sleep and quiet wakefulness, but they also can occur during behavior, albeit much less frequently.

We thank the reviewer's comment. We have rephrased to “SWRs are high-frequency oscillations (80-200Hz) in the local field potential (LFP) in the hippocampus that occur primarily during sleep or quiet wakefulness” (Page 3, Line 74)

3) Line 73: "SWs represent a mechanism for reactivating or consolidating…." Although evidence in favor of this idea continues to accumulate, I don't think it is proven yet, and the words "are thought to" should appear before "represent".

We appreciate the reviewer’s comment. We agree that further experimentation and analysis is needed to prove whether SWs represent a mechanism for reactivating or consolidating recent experiences. Now we have rephrased the sentence to “SWRs are thought to represent a mechanism for reactivating or consolidating recent experiences,”(Page 3 , Line 76).

4) Line 75: CA1 also outputs to the neocortex via direct projections back to deep layers of the EC

We appreciate the reviewer’s comment. We have rephrased to “~ allowing the ‘replayed’ information to be transferred from the CA3 region forward to the neocortex via the subiculum or deep layers of the entorhinal cortex as well as backward to the DG”. (Page 3, Line 77)

5) Line 80-81: The pioneering studies by Wilson & McNaughton (1994) and by Skaggs and McNaughton (1996) should be cited here.

We appreciate the reviewer’s suggestion. It is reflected and cited in the text. (Page 4, Line 92)

6) Line 105: Do the authors mean "included" here? Why would you exclude cells with thorny excrescences on the dendrites?

We apologize for the confusing text. We have rephrased the sentence to “Data were included in the analysis if the recorded cells had a membrane capacitance higher than 45 pF and had spines and thorny excrescences on their proximal dendrites ,” (Page 20, Line 631).

7) Line 209: What does prediction "deflection" mean? This term is not defined.

Sorry for the confusing wording. We have rephrased “Prediction deflection” as “Biased prediction” meaning that there is a predictable SWR bias within each MC. (Page 8, Line 233)

8) Line 467: What was the purpose of exposing the mice to an enriched environment? I assume this was for a different study, but it should be clarified.

We apologize for not providing a reason for exposing mice to an enriched environment. A similar question was asked by reviewer #1 and is answered above.

9) Figure S4 caption: The wording of this caption is confusing: "the SW of in vivo has a longer SW than that of in vitro"? Delete "that of"?

We apologize for the confusing text. We have rephrased the sentence to “the SWR of in vivo has a longer SWR than in vitro”.

[Editors’ note: what follows is the authors’ response to the second round of review.]

The manuscript has been improved but there are some remaining issues that need to be addressed, as outlined below:Reviewer #1 (Recommendations for the authors):The authors showed in their responses that they considered the concerns and took time to try to address the concerns. The efforts are appreciated. The manuscript is improved.However, several of the concerns were not well addressed.One issue that may not have been clear is that an explanation of the methods to construct a model of the network and the methods to use the model to develop the conclusions are not clear. This is critical so that the readers understand how the data were used to lead to the conclusions. In parts of the response some information is provided and this needs to be added to the manuscript. In addition, the additional methods need to be provided.

First, we added a new Method section called “Code Availability” to enable reproduction. Here, you can access URLs where you can find the data we used for analysis and details of the model (Page 27, Line 852). We have already described the model in detail in the section titled “Prediction of SWR waveforms *V*ms and the model architecture,” so please refer to that section for more information. (Page 24, Line 745)

Based on the comments from the reviewer, we added the following text to the Discussion section, focusing on how the RMSE results can be interpreted and lead to conclusion:

“To evaluate the predictive performance of the model, we used RMSE as a key metric. We found that the RMSEs for predictions using real data were significantly lower than those obtained using shuffled data, demonstrating that MC *V*ms contain structured, SWR-specific information. Had no difference been observed, it would suggest that the MC activity might only reflect nonspecific background input rather than meaningful information about individual SWRs. Similar results were also observed in the single MC analysis. We then focused on SWRs for which predictions based on real data had significantly lower RMSEs than those from shuffled data, and defined these as “predictable SWRs”— that is, SWR events whose information is encoded in the *V*m dynamics of a given MC. Therefore, we quantified the proportion of SWRs that each MC could predict. Thus, we applied RMSE not only as a measure of prediction accuracy, but also as a quantitative proxy for the amount and diversity of SWR-related information embedded in the subthreshold activity of MCs.” (Page 12, Line 346)

Some other fundamental concerns are raised by what the authors mention in their response:

Regarding (1) as described in the Methods section, the data criteria were clearly defined. “The data criterion for series resistance before and after recording was < 30%. The data were adopted when the mean resting membrane potential was < -50 mV and when Z-scores of the mean membrane potentials for 30 s were between -2 and 2. (Page 22, Line 684)” We consider cells to be healthy when these criteria are cleared. The start and end of the recording period are at the experimenter's discretion. For example, after recording a single dataset for 55 seconds, we confirm that the cell is healthy and then resume recording. If the cell remains healthy, we record again, and during this period, the recording electrode remains continuously inserted into the cell. Therefore, prolonged electrode insertion time may potentially impair cell health. We continue recording while periodically assessing cell health and adopting the data accordingly. As mentioned in the main text, the median recording time is 301 seconds, with recordings primarily aimed at five minutes. During this time, cell checks are performed periodically. While 55 seconds is a relatively short recording time, in this case, the recording was accidentally stopped due to an operational error. However, since there were no abnormalities in the cell's condition before or after the recording and a sufficient number of SWRs were recorded, we included the data. It is difficult to record SWRs multiple times in such slice experiments, but since machine learning requires a large number of SWRs, we include data with shorter recording times in the dataset. The points explained above are summarized in supplementary file 1. Another minor correction: the shortest time was not 55 seconds, but 56 seconds, so we have corrected this.

Regarding (2), all recorded cells were stained with biocytin. Only cells determined to be MCs by electrophysiological and histological criteria were included in the data. Some putative MCs that could not be stained were excluded. And we have already described it in the Methods section. “Data were included in the analysis if the recorded cells had a membrane capacitance higher than 45 pF (Hedrick et al., 2017) and had spines and thorny excrescences on their proximal dendrites (Murakawa and Kosaka, 2001; Amaral, 1978; Frotscher et al., 1991), which are electrophysiological and morphological markers of MCs. ” (Page 22, Line 686)

If MCs were recorded for 55-367sec, how could the authors know (1) the cell was healthy, and (2) the cell was a MC? Regarding (1), how did the authors assess the health and stability of the recording? It would seem they could not so they simply assumed it was healthy, despite only keeping the recording for a short time which one would think implies the cell was not healthy. That should be stated in the methods, one would think. Regarding (2), it is stated that cells were filled with biocytin, and one is shown. If no time was available to determine if the cell was a MC by physiology, then the biocytin is critical to identify that the cell was a MC. Was it the case that all cells were identified as MCs by biocytin, or was it only a subset? If some cells were not stained or not stained well enough, why were they included? There are many large cells in the hilus that are not MCs so just the size of the soma is not a good way to tell if a hilar cell is a MC.If MC depolarizations could not be analyzed by AUC or duration because there were barrages, this would be consistent with hilar cell physiology shown previously. The problem is that if barrages occur, one does not know if a SPWR caused a depolarization very well. This is because a SPWR has a duration that is 50-200 msec and barrages of EPSPs during that time will make it seem like MC depolarizations were caused by a SPWR but they actually occurred independently. A great way to make this easier is to use DCG-IV to stop the majority of MC spontaneous activity because it stops the GC input to MCs. If the authors do not choose to do this, then they need to make a better case that SPWRs induce MC depolarizations.

We have already demonstrated that EPSPs associated with SWRs disappear when the CA3 region and hilus region are dissected prior to recording (Figure 1—figure supplement 3). This clearly establishes that the EPSPs observed in the MC are derived from SWRs. Our experimental data suggest that synaptic transmission to the MC via mossy fibers is not affected by SWRs.

I apologize for not having read the Impact statement earlier. The current impact statement is quite complex, and the main message is not clear. Please revise.

We apologize for the complexity and confusion. We have rephrased the sentence to “Machine-learning algorithms combined with whole-cell patch-clamp and LFP recordings allow for reconstructing approximately 30% of SWR waveforms from the synaptic response of only five MCs.”

There are several assumptions that are not clearly based on data in the literature1. CA3 SPW-Rs are transmitted to the granule cells (GCs) via MCs. CA3 projects to both MCs and hilar GABAergic neurons. Together they innervate GCs, as well as each other. Therefore SPW-Rs in GCs are not necessarily a product of CA3 excitation of MCs alone. This was mentioned before and the authors added references that were suggested and content, which is good. But they added it in different locations. The Introduction still reads as if CA3 innervates MCs and MCs excite GCs, producing GC SPWRs. This is an oversimplification. Also, note that if the algorithm does not take into account the circuitry correctly it could make it seem like MCs are responsible for the events they call GC SPW-Rs and MCs are not solely responsible.

We have never mentioned GC SWR, but rather that synaptic responses to SWR were observed in GC. However, to avoid possible confusion, we have revised the text as follows:

“The observation of delayed synaptic responses in GCs compared to MCs during SWRs supports the hypothesis that the CA3-DG backpropagation reflects SWR-associated excitatory activity mediated by a disynaptic pathway involving MCs (Ouchi et al., 2017; Swaminathan et al., 2018).”(Page 4, Line 90)

2. Please note it also is not clear to this reviewer that LFPs like SPWRs occur in the GC layer. What is it that the authors think are "SPWRs" in recordings of the GC layer? Depolarizations? Dentate spikes?

In this study, LFP recordings from the GC layer were not performed. Furthermore, SWR was not mentioned as occurring in the GC layer.

3. It is confusing if the term SWR is referring to the ripple or the SPW or both. This is because SWR is defined as a ripple 80-200 Hz in one location of the paper. Furthermore, if it is, how would MCs produce a ripple in GCs if GABAergic neurons are not invoked? This seems to be a major point of confusion.

In response to previous comments from the reviewers, this paper uses the term "sharp wave ripples" (SWRs) to refer to both sharp waves and sharp wave ripples. The difference between sharp wave and ripple has been added to the Introduction. (Page 3, Line 76)

4. What is the reason why "MC ensembles" are assumed? It seems there is no basis for this. MCs are not interconnected, or are not interconnected much. Perhaps the term "ensembles" is not what the authors mean?

Since our paper mentions that multiple MC membrane potentials that are recorded simultaneously can carry diversity in SWR-related information, we believe that the term “ensemble” is conceptually appropriate. However, as reviewers pointed out, it is difficult to assert functional coupling, so we changed the term from “MC ensemble” to “a population of MCs”.

On line 74-76, the references that are cited did not show that SPWRs travel from CA3 to the DG by MCs. Pentonnen et al. did not prove MCs were critical. It was merely suggested. Scharfman did not either.

Yes, your comment is correct. The text has been revised overall in accordance with the comments of the reviewer, but please note that we wrote, "SWRs may also propagate backward to the DG"(Page 4, Line 86) so that we are also merely referring to a possibility.

Please explain the reasoning in the Introduction more clearly. For example, when are the authors discussing CA3 neurons and when are they discussion other neurons. When are they discussing the ripple and when are they discussing the sharp wave?

Based on the reviewer's comments, we have revised the relevant sections as follows:

“Sharp wave-ripples (SWRs) consist of two distinct components: the sharp wave and the ripple. The sharp wave is a slow deflection in the local field potential (LFP), primarily originating from the synchronous burst firing of CA3 pyramidal neurons. The ripple is a high-frequency oscillation (80–200 Hz) that is most prominently observed in the CA1 region, though it is also present in the CA3 region. The ripple component is superimposed on the sharp wave. SWRs primarily occur during sleep or quiet wakefulness (Buzsáki et al., 1992) and are thought to represent a mechanism for reactivating or consolidating recent experiences. These ‘replayed’ neural events allow information to be transferred from the CA3 region forward to the neocortex via the subiculum (Nitzan et al., 2020; Siapas and Wilson, 1998; Sirota et al., 2003) or to the deep layers of the entorhinal cortex (Chrobak and Buzsáki, 1994). SWRs may also propagate backward to the DG (Penttonen et al., 1997; Scharfman, 2007). In this case, SWRs generated by the population activity of CA3 pyramidal cells may propagate to the DG via synaptic inputs to MCs, which in turn provide excitatory input to GCs, forming a potential disynaptic pathway. The observation of delayed synaptic responses in GCs compared to MCs during SWRs supports the hypothesis that the CA3-DG backpropagation reflects SWR-associated excitatory activity mediated through MCs (Ouchi et al., 2017; Swaminathan et al., 2018). Furthermore, both experimental and theoretical studies have demonstrated that the MC activity is involved in memory processes such as spatial coding and pattern separation (Danielson et al., 2017; GoodSmith et al., 2017; Senzai and Buzsáki, 2017; Myers and Scharfman, 2011). In addition, MC synaptic activity serves as temporary storage and encoding of information at the local circuit level (Hyde and Strowbridge, 2012). In addition, MC synaptic activity serves as temporary storage and encoding of information at the local circuit level (Hyde and Strowbridge, 2012). ”(Page 3, Line 76)

Please explain the meaning of overcapacity on line 424. Why do the authors state that MCs form a layer? They are heterogeneously distributed. Throughout the Discussion it seems that the authors are making statements of fact based on their network model and this is not appropriate. In the model, for example, if the make MCs a layer, that does not mean they are in vivo. Please specify in the Discussion that the results are what the neural network found, not the data (especially lines 418 and following)

In the Introduction section of the main text, it is mentioned that the MC cell population is considered to be a relay point in the transmission pathway from CA3 to DG, but due to its relatively small cell number compared to other areas, it is considered a good experimental model for a bottleneck middle layer. Therefore, as you pointed out, although it is anatomically somewhat inaccurate, we used the term “MC layer.” To avoid confusion, we have revised the relevant section as follows:

“Thus, it is unclear how hippocampal information is efficiently encoded despite the limited capacity of the MC population.” (Page 15, Line 449)

Lines 434-437 are not clear. What the authors say they show is that CA3 has robust excitatory input to MCs, but this has been shown. Furthermore the authors seem to say they show MC activity occurs during SPWRs, but that has also been shown. On the other hand, one study did not find MC activity during SPWRs were not so consistent (Swaminathan et al.). The differences from the current study are not discussed.

We apologize for the lack of clarity. We are arguing that there is little evidence of a correlation between SWR and in vivo MC, and this context is unrelated to in vitro. To clarify the results of this study, we have revised the text as follows.

“However, in vivo experiments have not extensively investigated the relationship between SWR and MC. The significance of the fact that the SWRs recorded from CA3 are distributed and encoded in MC populations…” (Page 15, Line 462)

Other points:The response to the first comment is not very strong. The argument that the study proved connections between CA3 and MCs that have not been shown before is incorrect. Perhaps the authors are saying something else?

Thank you again for your valuable feedback. We fully agree that the anatomical projections from CA3 pyramidal neurons to mossy cells (MCs) are already well established in the literature, and we apologize if our previous response suggested otherwise. Our aim was not to claim novel anatomical connectivity, but rather to highlight a physiological demonstration of functional heterogeneity in how CA3 SWR activity is represented across the MC population. Specifically, our findings show that individual MCs exhibit distinct subthreshold membrane potential dynamics in response to the same SWR. Using machine learning-based decoding, we further demonstrate that different MCs carry only marginally overlapping information about CA3 SWRs. Notably, our results show that as the number of recorded MCs increases, up to at least five, the amount of SWR-related information that can be extracted grows progressively, indicating distributed and complementary encoding among MCs. This functional divergence, while anatomically plausible, had not been directly quantified or physiologically validated at the single-cell level during spontaneous SWRs. This is because no previous studies have simultaneously recorded the threshold potentials of multiple MCs, and thus the heterogeneity of MC synaptic responses to the same SWR has not been investigated. Therefore, we believe that this study provides quantitative insights into the mechanisms by which known anatomical backprojections generate functionally distributed encoding within MC populations.

In response to the second point the authors say their study shows the extent to which SWRs are encoded in MCs. I do not see the authors showed this. That conclusion is based on a very simplified model and methods that they do not describe well.

We agree that our model is relatively simple in its structure, using an encoder-decoder network. However, we believe that this simplicity adds value in a different sense. That is, if even a relatively simple model can predict aspects of SWR waveforms from MC Vms significantly better than chance, then it demonstrates that MCs do encode at least this level of information. And, we believe that more precise methods could potentially extract more information than our simple model.

This part is closely related to the reviewer's first comment, so please refer to the relevant section.

Explanations in the response are not always added to the paper. Please do so.

We incorporated as many of the previous comments as possible into the revised manuscript.

Reviewer #3 (Recommendations for the authors):The authors have addressed most of my points, but one of the major points is still unresolved.

We appreciate the reviewer’s critical evaluations, which have encouraged us to revise and resubmit this manuscript. We have changed the content as much as possible according to your suggestions.

Original comment 4: Although a mixed-effects model is preferable, the authors' new analyses are sufficient to show the same point, and this comment is considered resolved.

Thank you for your insightful comment. This comment greatly improved our manuscript.

Original comment 5: The author's response is unsatisfactory. They acknowledge that the problem I raised was correct, but they continue to show the result of this flawed statistical analysis in Figure 3C. Instead, they add a new supplementary figure with a paired t test (the rebuttal letter and the main text (line 179) state that this is Figure 3-supp Figure 5, but I believe it is really Figure 3-supp Figure 1, as supp Figure 5 does not exist). They need to remove Figure 3C entirely from the paper, as it is an inappropriate test for the reasons described in the last review (and which the authors acknowledge). Instead, they should put Figure 3-supp Figure 1 in its place.

We apologize for the discrepancies in responding to your comment. Figure 3C and E have been replaced with Figure 3-supp Figure 5 from the previous manuscript. Accordingly, changes have been made to the legend of Figure 3 and the corresponding result section in the text.

Original comment 6: The new discussion is a good addition. The authors should add Senzai and Buzsaki (2017) PMID: 28132824 and GoodSmith et al. (2017) PMID: 28132828 in the list of references with Bui et al. and Huang et al., as these papers were the first to show clearly that identified mossy cells had spatial tuning.

Thank you for the suggestion. We have cited further references (Page 15, Line 455).

[Editors’ note: what follows is the authors’ response to the third round of review.]

The manuscript has been substantially improved by your efforts but there are some remaining issues that need to be addressed, which can be summarized as follows:1. Limitations about the identification of mossy cells should be noted. 2. Consider adding experiments with DCG-IV. 3. Revise the significance statement for clarity. Detailed recommendations are below.Reviewer #1 (Recommendations for the authors):Again, the authors have improved the paper with their revisions.These residual concerns remain:1. Identifying mossy cells electrophysiologically and morphologically.The authors say that cells that "…had a membrane capacitance higher than 45 pF and spines and thorny excrescences on proximal dendrites …are electrophysiological and morphological markers of MCs."A cell capacitance over 45 pF is insufficient to define MCs electrophysiologically because it is also a characteristic of other cells. Spines do not characterize MCs because they are present on other cell types. Thorny excrescences do but all cells were not filled by biocytin (and we don't know how many were out of all those included).Therefore, please add a limitation to the text that the approach may have included other cell types, but the authors do not think this is a serious concern because the cells they filled showed thorny excrescences.

We are thankful for the reviewer’s suggestion. We have added following sentences:

“Since electrophysiological characteristics may include other cell types, we first confirmed the presence of thorny excrescences in the proximal dendrites using biocytin filling during recording. Only cells that met both criteria were classified as MCs and included in the analysis.” (Page 22, Line 693)

2. Regarding the evidence that CA3 SPWRs cause MC depolarizationsThe authors say that they have shown that synaptic transmission to the MC via mossy fibers in not affected by SPWRs. However, that was not the point of the suggestion. The point was that blocking mossy fiber transmission is a useful tool to reduce the non-CA3 input. If experiments were done in this condition, it would be more convincing if a SPWR caused a depolarization in MCs because the input that is not from CA3, the majority, would be removed.

Thanks for your comment. Our data suggest that CA3 SWRs propagate to MCs as synaptic inputs. However, the reviewer suggests that these MC EPSPs do not originate solely from SWR input, but rather from input via GCs in the DG.

Since barrage activity is also observed in MC EPSPs during SWRs, input via GCs through mossy fibers cannot be ruled out.

We will clearly describe the above points in the Discussion of the main text.

Our claim asserts that there is input from SWRs to MCs via backprojection and that SWR can be reconstructed from the corresponding EPSPs. Even if input from DG were superimposed, we believe it would not significantly impact the paper's claim. To support this assertion, we would like to organize the reviewer's concerns and our paper's claim as follows:

As the reviewer pointed out, when input from the GC is involved in the MC EPSP via SWR, anatomically, the following two cases are conceivable.

**Author response image 1. sa2fig1:** (A) CA3 SWRs propagate backward simultaneously to the MC and GC, with the input being sent to the MC (Diagram A). In this case, inhibiting input from the GC to the MC via the DCG-IV pathway may allow comparison of the magnitudes of the CA3-to-MC and GC-to-MC inputs. However, our previous studies that recorded EPSPs in the MC or GC during CA3 SWR showed that the SWR-to-MC pathway is significantly faster than the SWR-to-GC pathway in terms of time lag (Modified Figure 1,3 from Ouchi et al., 2017). This implies that, when a barrage of MC EPSPs occurs due to a SWR, the fast component likely originates from direct input from CA3 to MC rather than via GC. Regarding the slow component, input from GC may be involved. However, based on the causal diagram A, both components originate from the CA3 SWR. Therefore, even if there were SWR driven input from the GC to the MC EPSP in this study, it would not significantly undermine the argument that the signal can reconstruct the SWR because it originates from the same source. (B) A case in which input from GC to MC occurs without being caused by the CA3 SWR (Diagram B). In Figure 1 —figure supplement 3, severing the connection between CA3 and the hilus eliminates the input from SWR to MC. Therefore, the non SWR driven input from GC to MC is considered to have little effect.

Based on the above, the DCG-IV experiment suggested by the reviewer can be considered a useful method for isolating the direct transmission of CA3 SWR to MC.This is achieved by suppressing input from the GC to the MC, thereby reducing non-CA3 input. However, our findings provide evidence that CA3 SWR are transmitted to MCs and are sufficiently encoded in their subthreshold responses to allow for reconstruction. Even in the presence of potential superimposed inputs from the GC, such additional activity would not negate the specific correspondence we observed between CA3 SWRs and MC EPSPs. Rather, the robustness of decoding performance despite such confounds further supports the specificity of the CA3 SWR to MC backprojection.

To clarify the paper, we have added the following sentences:

“However, an important consideration in interpreting this study is that the resting membrane potential of MCs is highly active even under normal conditions, often exhibiting barrage activity where EPSPs accumulate immediately after their generation (Scharfman, 1993). Barrages were occasionally observed in MC EPSPs used in this study during SWRs. Therefore, the possibility that these EPSPs included input from GCs cannot be ruled out. Thus, it should be noted that the MC EPSPs used for machine learning do not necessarily reflect input purely from CA3 SWR. Nevertheless, the robustness of decoding performance despite such confounds further supports the specificity of the CA3 SWR to MC backprojection” (Page 14, Line 418).

3. Regarding the Impact statement that was unclearIt is still unclear. Why is it important that 30% of SWR waveforms were reconstructed. How would a synaptic response of a MC reconstruct a SPR? How would 5 MCs do this?Is this the meaning? Machine-learning algorithms combined with whole-cell patch-clamp of MCs and LFP recordings of SWRs showed MC EPSPs follow approximately 30% of SWRs?

We apologize for any confusion this may have caused. The previous impact statement could have been interpreted as the reconstruction of 30% of a single SWR waveform.

As shown in Figure 3A, SWR can be reconstructed based on the synaptic responses of MCs corresponding to the SWR. We then analyzed the proportion of the entire SWR that could be reconstructed by a single MC. Using the RMSE metric, we determined that SWR was significantly reconstructed when the prediction based on real data showed a significantly lower RMSE than the prediction based on shuffled data. A single MC could significantly reconstruct approximately 10% of the total of SWR. Furthermore, calculating the proportion of the SWR that was significantly reconstructed by each of the five simultaneously recorded MCs revealed that they collectively significantly reconstructed approximately 30% of the recorded SWR, despite some overlap (Figure 6B, C).

Despite the MC having a far smaller number of cells (30,000) compared to the CA3 region (300,000), we were surprised that approximately 30% of the total SWR could be reconstructed from the synaptic responses of just five MCs. We consider this an important finding, as it indicates that precise information is encoded within the synaptic responses of MCs.

Therefore, we revised the text as follows.

“Combining machine learning with whole-cell patch-clamp and LFP recordings reveals that synaptic responses from only five mossy cells, despite their limited population, can significantly reconstruct about 30% of the total hippocampal sharp wave-ripples, highlighting efficient distributed coding within subthreshold activities.”

4. Regarding the response starting with "We have never mentioned GC SWR…" The authors actually revised to say what this reviewer meant, so that is good. One minor point, which is about the references in the text the authors revised. It is that the back projection that is described in the sentence was originally shown by PMIDs 7931561, 7884451, 9157312 and evidence reviewed in PMID 17765742.

Thank you for your suggestion. We have cited further references (Page 4, Line 92).